# A widespread family of serine/threonine protein phosphatases shares a common regulatory switch with proteasomal proteases

Niels Bradshaw[1], Vladimir M Levdikov[2], Christina M Zimanyi[1†], Rachelle Gaudet[1], Anthony J Wilkinson[2], Richard Losick[1*]

[1]Department of Molecular and Cellular Biology, Harvard University, Cambridge, United States; [2]Structural Biology Laboratory, Department of Chemistry, University of York, York, United Kingdom

**Abstract** PP2C phosphatases control biological processes including stress responses, development, and cell division in all kingdoms of life. Diverse regulatory domains adapt PP2C phosphatases to specific functions, but how these domains control phosphatase activity was unknown. We present structures representing active and inactive states of the PP2C phosphatase SpoIIE from *Bacillus subtilis*. Based on structural analyses and genetic and biochemical experiments, we identify an $\alpha$-helical switch that shifts a carbonyl oxygen into the active site to coordinate a metal cofactor. Our analysis indicates that this switch is widely conserved among PP2C family members, serving as a platform to control phosphatase activity in response to diverse inputs. Remarkably, the switch is shared with proteasomal proteases, which we identify as evolutionary and structural relatives of PP2C phosphatases. Although these proteases use an unrelated catalytic mechanism, rotation of equivalent helices controls protease activity by movement of the equivalent carbonyl oxygen into the active site.

*For correspondence: losick@mcb.harvard.edu

Present address: †New York Structural Biology Center, New York, United States

Competing interests: The authors declare that no competing interests exist.

## Introduction

Reversible protein phosphorylation is widely used in biological systems to control the activity of enzymes or the association of proteins with other proteins. Kinases and phosphatases control the phosphorylation state of target proteins in response to specific cellular or environmental cues, making reversible phosphorylation a flexible mechanism to control diverse biological systems (*Huse and Kuriyan, 2002*; *Shi, 2009*; *Taylor and Kornev, 2011*). Here we address the question of how members of the PP2C family of serine/threonine phosphatases are regulated to control processes such as cell growth and death, development, and responses to stress in all kingdoms of life (*Kerk et al., 2015*; *Lammers and Lavi, 2007*; *Shi, 2009*). Among serine/threonine phosphatases, a distinctive feature of the PP2C family is that the activity of a conserved catalytic domain is controlled by diverse regulatory domains that are often linked in cis to the catalytic domain (*Shi, 2009*; *Zhang and Shi, 2004*). We investigated the PP2C family member SpoIIE, which controls the activation of the cell-specific transcription factor $\sigma^F$ during the developmental process of sporulation in the bacterium *Bacillus subtilis*.

Sporulation involves the formation of an asymmetrically-positioned septum that divides the developing cell into large and small cellular compartments (*Stragier and Losick, 1996*). SpoIIE is the most upstream member of a three-protein pathway that activates $\sigma^F$ in the small cell (*Figure 1A*). It does so by dephosphorylating the phosphoprotein SpoIIAA-P (*Duncan et al., 1995*).

**eLife digest** To regulate the activity of proteins, cells often modify them by adding or removing chemical groups called phosphates. Therefore, the enzymes that add or remove these phosphate groups must be tightly regulated so that they are active at the right time and place. Enzymes known as phosphatases remove phosphate groups from proteins and the PP2Cs are one such family of enzymes that are found in bacteria, plants and animals. Despite their broad importance, it was not clear how cells control the PP2Cs.

One way to understand how an enzyme is controlled is to compare the three-dimensional structures of the enzyme when it is active and when it is inactive. Bradshaw et al. used a PP2C enzyme from bacteria as a model to understand how the cell regulates other PP2Cs.

The experiments reveal that the bacterial enzyme has a structural element that acts as a switch to control its activity. The phosphatase needs to bind metal ions to be active, and movement of the switch promotes binding of the metal ions to activate the phosphatase. The switch is also found in other members of the PP2C family. Furthermore, members of a seemingly unrelated family of enzymes called the proteasomal proteases, which degrade proteins, also have a similar architecture and are controlled by a similar switch. Thus, the phosphatase and protease families may have a common evolutionary history.

Multiple members of the PP2C family are involved in cancer and other diseases. The discovery of a regulatory switch provides new opportunities to use drugs to control phosphatase activity in patients. Many cancer drugs that are currently in use or are under development target enzymes that add phosphate groups to proteins, but efforts to target the phosphatases have largely been unsuccessful. Bradshaw et al.'s findings may enable the development of new drugs that target protein phosphatases.

Dephosphorylated SpoIIAA, in turn, displaces $\sigma^F$ from the anti-sigma factor SpoIIAB to release the free and active transcription factor (*Figure 1A*) (*Diederich et al., 1994*). A long-standing mystery is how SpoIIE is regulated to generate dephosphorylated SpoIIAA selectively in the small cell. Recent work indicates that SpoIIE initially associates with the asymmetrically-positioned cytokinetic ring and then during cytokinesis is handed off to the adjacent cell pole, which will become the small cell (*Bradshaw and Losick, 2015*). Cell-specific activation is mediated by the self-association of SpoIIE molecules in the small cell, which protects the protein from proteolysis and activates the phosphatase (*Bradshaw and Losick, 2015*). Here we focus on the molecular mechanism of phosphatase activation.

Like other PP2C family phosphatases, the catalytic center of SpoIIE uses two divalent cations (manganese in the case of SpoIIE) to deprotonate a water molecule that serves as the nucleophile for dephosphorylation (*Arigoni et al., 1996*; *Schroeter et al., 1999*). This active site is embedded in the conserved fold of the PP2C domain, which is shared by all PP2C family members (*Shi, 2009*). The PP2C domain is paired with diverse regulatory modules (over 1500 unique domain architectures have been identified in the InterPro database) (*Mitchell et al., 2015*), but how these regulatory modules control phosphatase activity was not understood. Here we identify a pair of $\alpha$-helices at the heart of the regulatory mechanism that rotate to position a carbonyl oxygen to bind an active site $Mn^{2+}$ ion and activate SpoIIE. We present evidence that this mechanism is widely conserved among PP2C family members. Remarkably, rotation of equivalent $\alpha$-helices is also used to control an unrelated catalytic mechanism in the structurally similar family of enzymes that form the catalytic core of the proteasome (*Arciniega et al., 2014*; *Ruschak and Kay, 2012*; *Shi and Kay, 2014*; *Sousa et al., 2000*). This raises the possibility that PP2C phosphatases and proteasome proteases have a common evolutionary history that is linked by a shared regulatory mechanism.

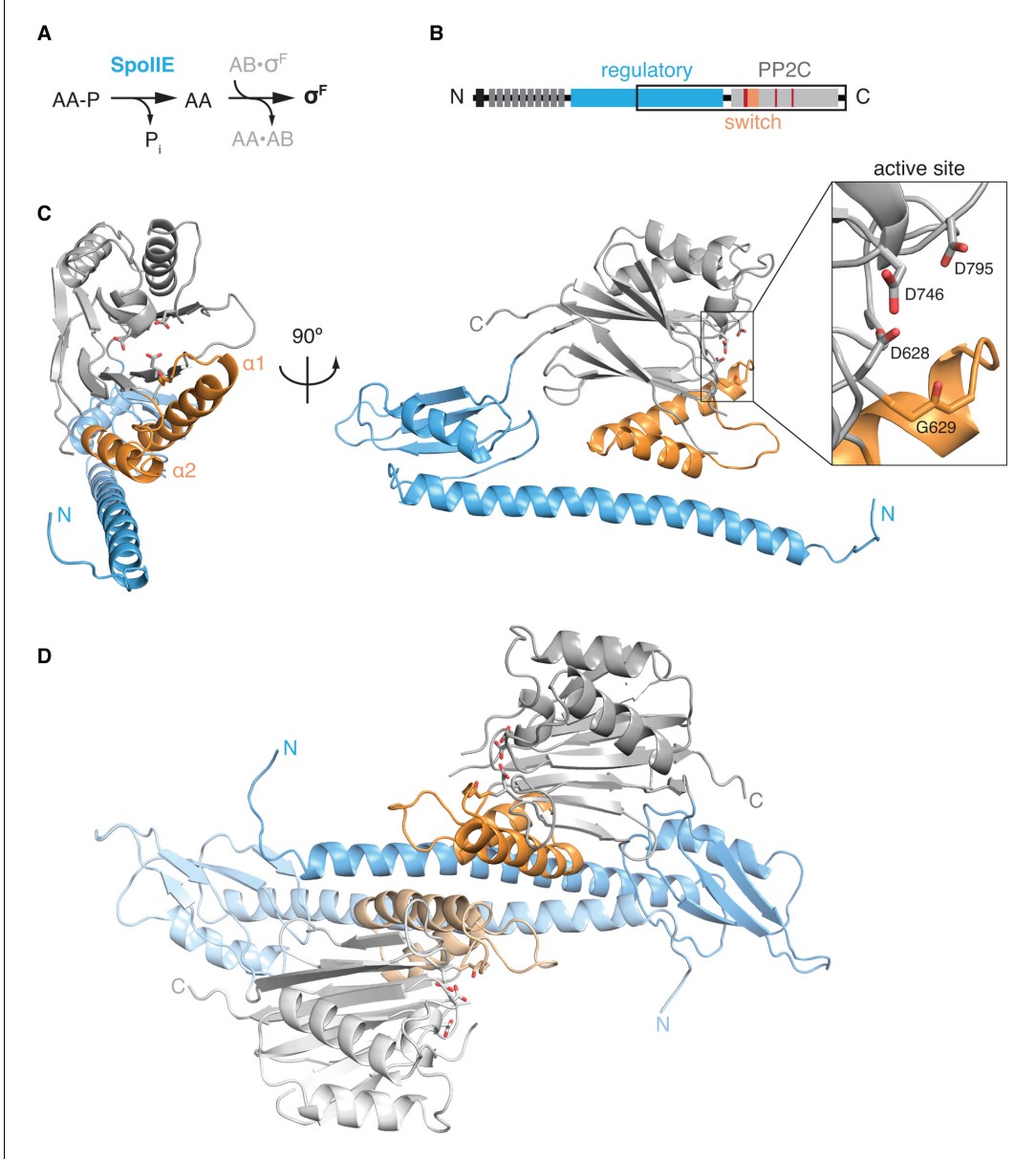

**Figure 1.** The structure of SpoIIE with its regulatory domain. **A** is a diagram of the three-protein pathway controlling $\sigma^F$. **B** is a schematic diagram of the SpoIIE primary structure with its N-terminal cytoplasmic degradation tag in black, the 10 transmembrane segments in dark grey, the regulatory domain in blue, and the PP2C phosphatase domain shown in light grey. Also shown are the switch helices in orange and the metal-coordinating residues within the active site in red. The black box identifies the SpoIIE$^{457-827}$ fragment that was crystallized. **C** is a ribbon diagram of a single molecule of SpoIIE$^{457-827}$ with front and side views using the same color scheme as the diagram in panel B. The inset shows the putative metal coordinating sidechains of the active site (from top to bottom: D795, D746, and D628) and the backbone carbonyl of G629. *Figure 1—figure supplement 1* shows the 2F$_o$-F$_c$ electron density map and a stereo representation of the SpoIIE$^{457-827}$ structure. **D** shows the dimer observed in the crystal structure of SpoIIE$^{457-827}$ (chains A and B) with the two protomers in darker and lighter shades (buried surface area 1500–2000 Å$^2$ per monomer). The two and a half dimers in the asymmetric unit are shown in *Figure 1—figure supplement 2*.

The following figure supplements are available for figure 1:

**Figure supplement 1.** Validation of the SpoIIE$^{457–827}$ structure.

**Figure supplement 2.** The crystal lattice contains three similar SpoIIE dimers.

# Results

## Overview

To investigate how PP2C phosphatase activity is regulated, we sought to determine X-ray crystal structures of SpoIIE with the phosphatase in the active and inactive states. We present a structure of a fragment that includes the entire PP2C phosphatase domain and a portion of the adjacent regulatory domain. This structure shows that the regulatory domain mediates the formation of dimers between SpoIIE molecules, and evidence indicates that dimerization is needed to activate the phosphatase. We also present a structure of the phosphatase domain alone. A comparison of the structures reveals that dimerization rotates two α-helices of the PP2C fold (α1 and α2 of the conserved PP2C fold) (*Das et al., 1996*) relative to the phosphatase core. We refer to these helices as switch helices and present evidence that this shift in position switches the phosphatase from the inactive to active state.

## Structure of the phosphatase domain with a portion of the adjacent regulatory domain

To determine how SpoIIE is regulated, we first sought to determine the structure of the molecule in an active, self-associated state. The entire, 270-residue-long regulatory domain mediated the formation of heterogeneous multimers that were refractory to crystallization (*Bradshaw and Losick, 2015*). Using bioinformatic analysis, we devised a construct (SpoIIE$^{457-827}$) that included the C-terminal half of the regulatory domain and the PP2C phosphatase domain (*Figure 1B*; information on the design of the construct is presented in the Materials and methods). This construct produced monodisperse protein that yielded crystals. Despite limited (3.9 Å) resolution of the diffraction data, the overall secondary structure elements were well-defined in electron density maps for both the regulatory and the phosphatase domains (*Figure 1*, *Figure 1—figure supplement 1*, and *Table 1*). The most striking feature of the regulatory domain was an N-terminal 45-residue long α-helix (residues 473–518) that makes intramolecular contacts with the switch helices (α1 and α2) of the phosphatase domain (*Figure 1C*).

The five molecules of SpoIIE$^{457-827}$ in the asymmetric unit were paired in similar dimers; two dimers were formed within the asymmetric unit and the fifth molecule dimerized across a crystallographic two-fold axis (*Figure 1D*, *Figure 1—figure supplement 2*). The core of the dimer interface (1500–2000 Å$^2$ buried surface per monomer) was formed from antiparallel contacts between the long α-helices from the regulatory domains of adjacent molecules. Additionally, the switch helices at the base of each phosphatase domain contact each other across the dimer interface (*Figure 1*, *Figure 2A and B*, shown in orange).

## Amino acid substitutions in the dimer interface block function

To investigate the role of dimerization in stabilization, localization and phosphatase activation, we systematically created substitutions of residues that make up the dimer interface and investigated the ability of these mutants to function during sporulation. We substituted the native amino-acids with lysine because the positive charge and the long side chain would be expected to impair dimerization. The effect of these substitutions on stabilization and subcellular localization was investigated by use of a SpoIIE-YFP fusion and the effect on phosphatase activity was judged by use of a σ$^F$-dependent reporter (*Figure 2B*, red circles, *Figure 2—figure supplement 1A,B and C*). The results revealed that a continuous region of the dimer interface (marked with red circles in *Figure 2B*) composed of six residues from the long α-helix of the regulatory domain (V480, L484, V487, M491, F494, and I498) and three residues from the switch helices (L646, I650, and T663) were needed for all three aspects of SpoIIE function. These findings are consistent with the hypothesis that the dimers observed in our structure represent the active state of the phosphatase.

## Structure of the phosphatase domain

To investigate how dimerization activates phosphatase activity, we sought to compare the active dimeric structure of SpoIIE$^{457-827}$ to inactive SpoIIE. Previously, we determined the structure of SpoIIE$^{590-827}$, a fragment that included the PP2C phosphatase domain but lacked the adjacent regulatory domain (*Levdikov et al., 2012*). We hypothesized that this structure represented the inactive

**Table 1.** Data collection and refinement statistics.

| | SpoIIE[457-827] (5UCG) | SpoIIE[590-827] (5MQH) |
|---|---|---|
| Data collection | | |
| Beam source | APS 24-ID-C | Diamond, I02 |
| Wavelength (Å) | 0.9792 | 0.97950 |
| Space group | $P4_32_12$ | $C222_1$ |
| Cell dimensions | | |
| *a*, *b*, *c* (Å) | 125.62, 125.62, 330.70 | 56.29, 122.51, 81.62 |
| α, β, γ (°) | 90, 90, 90 | 90, 90, 90 |
| Resolution (Å)* | 60–3.9 (3.97–3.9) | 61.34–2.44 (2.48–2.44) |
| Total reflections* | 284918 (8031) | 60359 (4228) |
| Unique reflections* | 24917 (1181) | 10961 (681) |
| $R_{sym}$[†]* | 0.102 (1.448) | 0.057 (0.631) |
| $CC_{1/2}$ | 0.999 (0.847) | 0.999 (0.874) |
| CC* | 1.00 (0.958) | - |
| $I / \sigma I$* | 24.7 (0.8) | 20.1 (2.8) |
| Completeness (%)* | 99.7 (97.4) | 99.7 (99.8) |
| Redundancy* | 11.4 (6.8) | 6.3 (6.2) |
| Refinement | | |
| Resolution (Å)* | 50–3.9 (4.1–3.9) | 50–2.45 (2.51–2.45) |
| No. reflections | 21558 | 10187 |
| $R_{work}$ / $R_{free}$[‡]* | 0.28/0.32 | 0.21/0.28 |
| No. atoms | | |
| Protein | 13166 | 1783 |
| *B*-factors | | |
| Protein | 93.0 | 68.0 |
| R.m.s. deviations | | |
| Bond lengths (Å) | 0.002 | 0.010 |
| Bond angles (°) | 0.525 | 1.545 |
| Ramachandran plot | | |
| Favored (%) | 92.48 | 96.9 |
| Allowed (%) | 7.40 | 3.1 |
| Outliers (%) | 0.12 | 0 |
| Rotamer outliers (%) | 6.44 | 15.4 |

*Values in parentheses are for highest-resolution shell.

[†]$R_{sym} = \sum_{hkl}\sum_i |I_i - <I>| / \sum_{hkl}\sum_i <I>$ where $I_i$ is the intensity of the *i*th measurement of a reflection with indexes *hkl* and $<I>$ is the statistically weighted average reflection intensity.

[‡]$R_{work} = \sum ||F_o| - |F_c|| / \sum |F_o|$ where $F_o$ and $F_c$ are the observed and calculated structure factor amplitudes, respectively. $R_{free}$ is the R-factor calculated with 5% of the reflections chosen at random and omitted from refinement.

state because it lacked the dimeric interface of the SpoIIE[457–827] structure. Although monomeric in solution under physiological conditions, SpoIIE[590–827] had undergone a domain-swap dimerization during crystallization (*Levdikov et al., 2012*). Here, we solved an additional structure for SpoIIE[590–827] (with an amino acid substitution A624I that was designed to block domain swapping) that was in a different crystal form and was not domain-swapped (*Figure 3A*). Importantly, the only significant differences between the two SpoIIE[590–827] structures were at the site of the domain-swap (*Figure 3—*

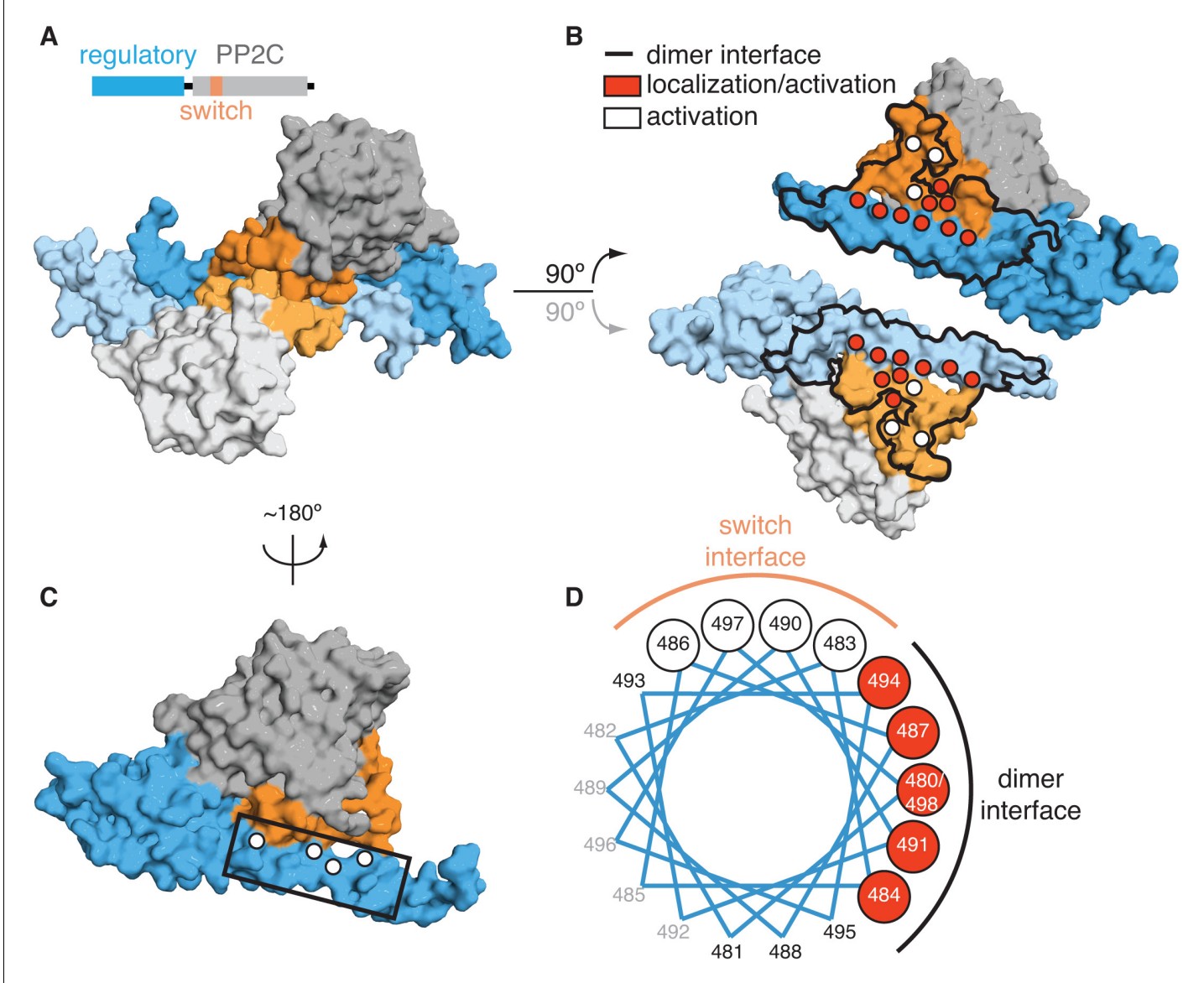

**Figure 2.** Dimerization activates the phosphatase. A is a surface representation of the SpoIIE$^{457-827}$ dimer with the phosphatase domain, the switch, and the regulatory domain color coded as indicated in the associated schematic. Chain A is colored with darker shades and Chain B is colored with lighter shades. B is an open-book view of the SpoIIE$^{457-827}$ dimer with the interface (defined as residues within 4.5 Å of the adjacent molecule) outlined in black. Red circles mark positions of amino-acid substitutions that blocked stabilization, localization, and activation (V480K, L484K, V487K, M491K, F494K, I498K, L646K, I650K, and T663K), whereas white circles mark positions of substitutions that blocked activation (as judged by σ$^F$ activity) but not stabilization and localization (E639K, E642K, and I667K). *Figure 2—figure supplement 1* presents the analysis of the behavior of the SpoIIE mutants *in vivo*. C is a surface representation of Chain A of SpoIIE$^{457-827}$ rotated approximately 180° relative to the dimeric view in A. White circles indicate positions of substitutions that led to defects in activation (but not localization) of SpoIIE in vivo (Q483A, G486K, V490K, and E497K). The box outlines the section of the long α-helix of the regulatory domain that is represented as a helical wheel in D. *Figure 2—figure supplement 1* presents the analysis of the behavior of the SpoIIE mutants *in vivo*. D is a helical wheel representation of residues 480 to 498 from the long α-helix of the regulatory domain. Positions at which substitutions led to defects in σ$^F$ activation are indicated by circles colored as in B and C. Black text (A481K, S488K, D493K, and S495K) indicates positions where substitutions did not lead to a phenotype, grey text represents positions that were not tested. *Figure 2—figure supplement 1* presents the analysis of the behavior of the SpoIIE mutants *in vivo*.

The following figure supplement is available for figure 2:

**Figure supplement 1.** Functional analysis of the dimer interface.

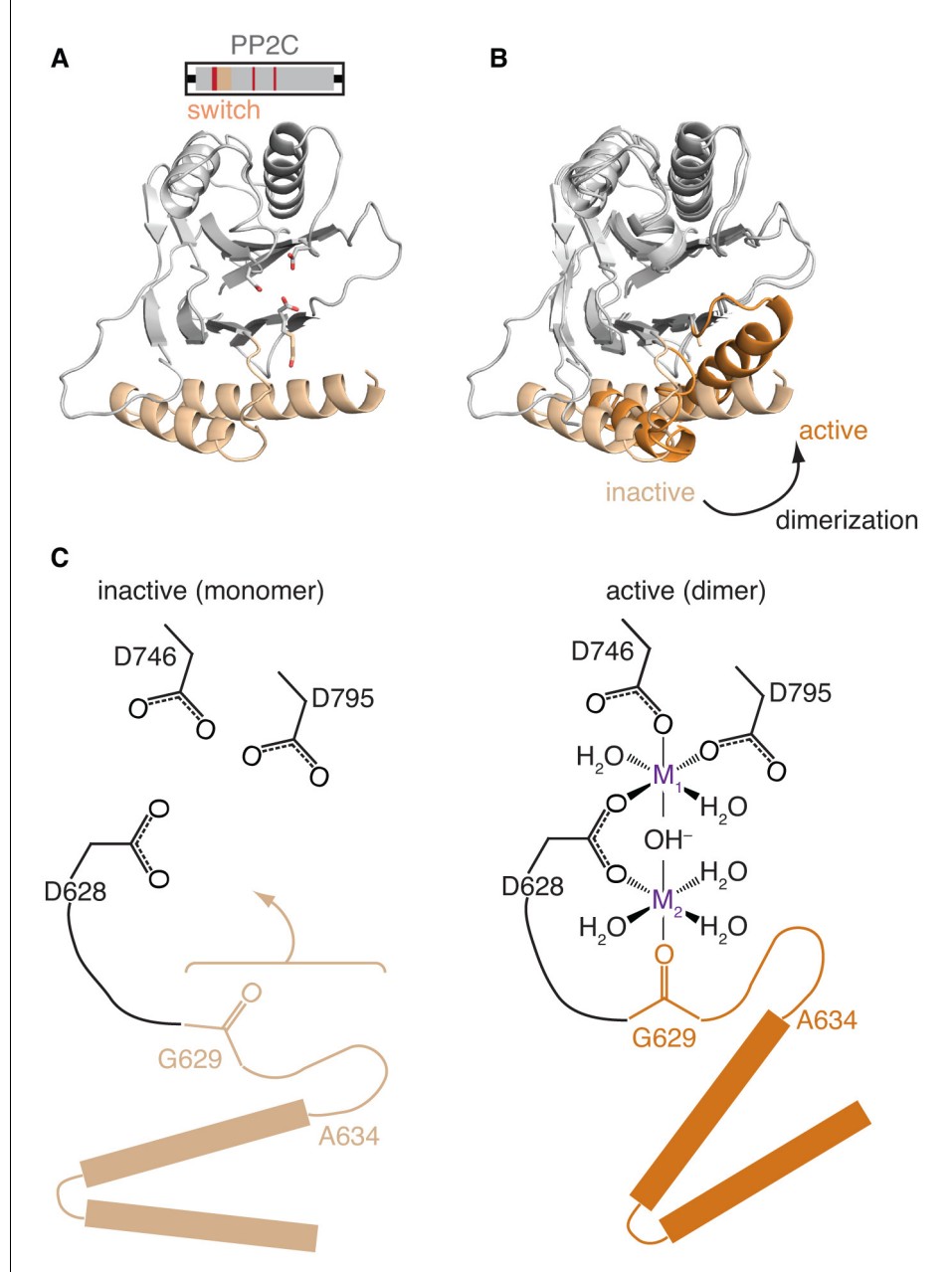

**Figure 3.** Repositioning the switch activates the phosphatase. **A** is a ribbon diagram of the structure of SpoIIE[590–827], which is the phosphatase domain of SpoIIE lacking the regulatory domain. The region of the protein that was crystallized is diagramed above. The switch region and $Mn^{2+}$-coordinating residues are color-coded as in *Figure 1A*. *Figure 3—figure supplement 1* shows a comparison with the previously published domain swapped SpoIIE[590–827] structure. **B** compares the conformations of the phosphatase domain in the dimeric SpoIIE[457–827] structure (switch helices in dark orange) and the isolated phosphatase domain of SpoIIE[590–827] (switch helices in light orange). The structures were aligned based on the core of the phosphatase domain excluding the switch region (residues 590–628 and 678–827) with an RMSD = 0.952 Å (970 to 970 atoms). The major conformational change upon dimerization corresponds to a rotation and upward movement of the switch helices. *Figure 3—figure supplement 2* shows how gain of function mutants may promote the conformational change. **C** is a model for how rotation of the switch helices leads to phosphatase activation. In the inactive state (left) G629 is not positioned to coordinate the M2 metal. We propose that dimerization (right) leads to rotation of the switch helices (orange), which repositions G629 to recruit manganese and complete the active site. We note that an additional glycine of RsbX (G47), corresponding to G631 of SpoIIE, also coordinates M2. Thus, it is possible that G631 also

*Figure 3 continued on next page*

*Figure 3 continued*

coordinates M2 in place of the lower right-hand water molecule depicted in the schematic diagram (*Teh et al., 2015*). *Figure 3—figure supplement 3* shows details of the active site in the SpoIIE[457–827] structure.

The following figure supplements are available for figure 3:

**Figure supplement 1.** Comparison of the SpoIIE[590–827] structures.

**Figure supplement 2.** Gain-of-function alleles activate the phosphatase.

**Figure supplement 3.** Manganese binding in the SpoIIE active site.

*figure supplement 1*). Also, contacts between the phosphatase domains observed in the SpoIIE[457–827] dimer were not present in either of the SpoIIE[590–827] structures.

Comparison of the SpoIIE[590–827] structures with SpoIIE[457–827] revealed that dimerization rotated the switch helices (α1 and α2 of the PP2C fold, corresponding to SpoIIE residues 630–678) approximately 45° as a rigid body relative to the phosphatase core (*Figure 3B*, *Video 1*). We hypothesized that this conformational change of the switch helices is responsible for activation of the SpoIIE phosphatase.

## Repositioning the switch region is necessary for phosphatase activation

To evaluate whether repositioning of the switch region is responsible for phosphatase activation, we returned to our genetic analysis of the contacts made in the SpoIIE[457–827] structure. In the dimer, the switch helices are held in position by intramolecular contacts with the long α-helix of the regulatory domain and intermolecular contacts between switch helices across the dimer interface (*Figures 1D*, *2B and C*). We found that single-amino acid substitutions at either of these contact sites blocked phosphatase activity but not stabilization or localization to the small cell. Phosphatase activity was assessed by σ[F]-directed gene expression and stabilization and localization by use of a SpoIIE-YFP fusion (white circles in *Figure 2C and D* and *Figure 2—figure supplement 1A and D*). This result defines two roles for the long α-helix: one face of the helix mediates dimerization and is required for all three aspects of SpoIIE function (stabilization, localization and phosphatase activity) (*Figure 2B and D* red circles), and the other face, which makes intramolecular contacts with the switch region, is specifically required for phosphatase activity (*Figure 2B–D* white circles). Additionally, these results are consistent with the idea that dimerization stimulates phosphatase activity by repositioning the switch helices.

## Evidence from gain-of-function mutants that repositioning the switch helices is sufficient for phosphatase activation

Replacement of valine at position 697 with alanine causes a gain-of-function mutant phenotype in which σ[F] is activated constitutively (*Carniol et al., 2004*; *Hilbert and Piggot, 2003*). The V697A substitution also enhanced phosphatase activity as measured in vitro (*Bradshaw and Losick, 2015*). But how this

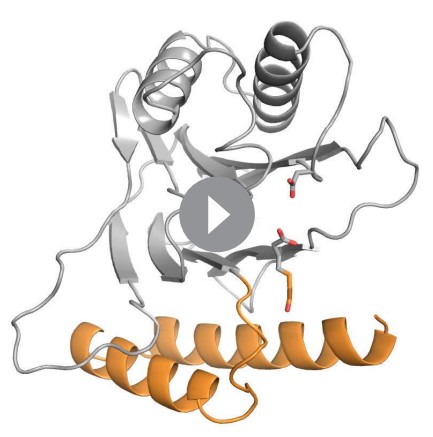

**Video 1.** The PP2C phosphatase domain of SpoIIE changes conformation upon dimerization. Shown is the PP2C phosphatase domain of SpoIIE (switch helices in orange) morphing from the structure of SpoIIE[590–827] to the structure of SpoIIE[457–827]. The structures were aligned based on the core of the phosphatase domain excluding the switch region (residues 590–628 and 678–827) as in *Figure 3B*.

substitution acts had been unclear. Our structure of SpoIIE$^{590–827}$ reveals that in the monomeric state, V697 packs in a hydrophobic pocket between the $\beta$ strands at the base of the PP2C domain and the switch (*Figure 3—figure supplement 2A*). In contrast, N665 from the switch packs near V697 in the structure of SpoIIE$^{457–827}$ in the dimeric state. We therefore hypothesize that in the wild type, V697 stabilizes the conformation of the switch helices in the inactive monomeric state and that truncating V697 to alanine stabilizes the active conformation by promoting solvation of the polar residue N665 (*Figure 3—figure supplement 2A*). Thus, and according to our hypothesis, replacing V697 with alanine destabilizes the inactive state by removing hydrophobic contacts and favors the active conformation by eliminating a repulsive interaction. Reinforcing this hypothesis, substitution of V697 with a bulky hydrophobic residue (phenylalanine), which could similarly destabilize the inactive conformation, also causes constitutive activity (*Bradshaw and Losick, 2015*).

Other gain-of-function mutants that stimulate phosphatase activity in the context of a loss-of-function mutant support this hypothesis (*Carniol et al., 2004*). The amino acid substitutions in these mutants (L479F, K649T, I650L, I684V, L695W, and V728M) were all located at positions in the structure that could contribute to positioning the switch helices (*Figure 3—figure supplement 2B*). I684 and L695 project down from the $\beta$-strands at the base of the phosphatase domain to contact the switch. K649 and I650 are themselves part of the switch helices and project across the dimer interface. V728 projects towards the switch from the loop implicated in substrate binding in other PP2C phosphatases. Finally, L479 projects up towards the switch from the long $\alpha$-helix of the regulatory domain. We conclude that, like V697A, these amino-acid substitutions bias the phosphatase domain to the active conformation of the switch region.

## The switch helices move a conserved manganese-coordinating residue into the active site

How does repositioning the switch region activate phosphatase activity? All PP2C phosphatases coordinate 2–3 divalent metals (usually manganese) in their active sites (*Das et al., 1996*; *Shi, 2009*). The two core metal ions, known as M1 and M2, directly participate in catalysis by deprotonating a water molecule that serves as the nucleophile for hydrolysis (*Das et al., 1996*). Based on the universally conserved architecture of the catalytic center, the M2 metal of SpoIIE is predicted to be coordinated by the side-chain of D628 and the carbonyl oxygen of G629 (*Schroeter et al., 1999*) (*Figure 1C* and *Figure 3C*). G629 is at the junction between the switch helices and the $\beta$ strands at the base of the phosphatase domain, such that movement of the switch helices could be coupled with bringing G629 into position to recruit M2.

In support of this idea, G629 is not in position to coordinate M2 in our isolated phosphatase domain structures, which we thus conclude represent an inactive state. This is supported by the fact that although our previously published structures included manganese in the crystallization conditions, the M2 site was unoccupied and the active site contained only a single manganese (*Levdikov et al., 2012*). While soaking SpoIIE$^{457–827}$ crystals with manganese degraded the diffraction, an anomalous difference map provided evidence that manganese was bound in the active site (*Figure 3—figure supplement 3A* and *Table 2*). Due to the low (5.4 Å) resolution of the data for the manganese-soaked crystals, the number of bound metal ions and their position in the active site could not be established. In the dimeric SpoIIE$^{457–827}$ structure and in contrast to the SpoIIE$^{590–827}$ structure, the loop connecting the switch helices to G629 was ordered (*Figure 3—figure supplement 3B*) and overlaid well with M2-containing structures of closely related phosphatases such as *B. subtilis* RsbX (*Teh et al., 2015*), *M. tuberculosis* Rv1364c (*King-Scott et al., 2011*), and *S. thermophilus* Sthe_0969 (*Nocek et al., 2010*) (*Figure 3—figure supplement 3C*). We propose that the shift of the switch helices activates the phosphatase by repositioning G629 to recruit M2 and complete the active site (*Figure 3C*).

## Mn$^{2+}$ stimulates dimerization and phosphatase activity

A prediction of the hypothesis that movement of the helices allows recruitment of M2 is that binding of metal to the active site of SpoIIE should be coupled to dimerization and activation. Whereas in cells, cues in the forespore promote self-association of SpoIIE to induce phosphatase activity (*Bradshaw and Losick, 2015*), we reasoned that in vitro, in the absence of cellular cues, addition of high concentrations of manganese should drive dimerization and activation by mass action

**Table 2.** Data collection statistics for anomalous datasets.

| | SpoIIE[457-827] Mn | SpoIIE[457-827] SeMet |
|---|---|---|
| **Data collection** | | |
| Beam source | APS 24-ID-C | APS 24-ID-E |
| Space group | $P4_32_12$ | $P4_32_12$ |
| Cell dimensions | | |
| $a, b, c$ (Å) | 124.783, 124.783, 329.787 | 123.081, 123.081, 329.556 |
| $\alpha, \beta, \gamma$ (°) | 90, 90, 90 | 90, 90, 90 |
| | *Inflection* | *Inflection* |
| Wavelength (Å) | 1.89350 | 0.97920 |
| Resolution (Å)* | 50–5.4 (5.49 — 5.4) | 50–5.7 (5.8–5.7) |
| Total reflections* | 40325 (318) | 51233 (4024) |
| Unique reflections* | 8598 (187) | 8071 (706) |
| $R_{sym}$* | 0.145 (0.535) | 0.175 (1.475) |
| $CC_{1/2}$* | 0.99 (0.75) | 0.996 (0.459) |
| CC* * | 0.997 (0.926) | 0.999 (0.793) |
| *Mean I / σI** | 9.14 (1.00) | 7.86 (1.13) |
| Completeness (%)* | 90.1 (41.6) | 99.0 (97.4) |
| Redundancy* | 4.7 (1.7) | 6.3 (5.7) |

*Values in parentheses are for highest-resolution shell.

(*Figure 4A*). We used size exclusion chromatography coupled to multi angle laser light scattering (SEC-MALLS) to monitor SpoIIE dimerization over a range of manganese concentrations. In the absence of manganese, SpoIIE[457–827] eluted as a single monodisperse peak with a calculated molecular weight of 42 kDa, consistent with the calculated molecular weight of a monomer (*Figure 4B*). Addition of 250 µM and 1 mM MnCl$_2$ induced dimerization of SpoIIE[457–827], shifting and broadening the peak in concert with an increase in molecular mass (*Figure 4B*). In support of the idea that this dimerization uses the interface found in the SpoIIE[457–827] structure, substitution of a residue from the interface (L484) with lysine blocked dimerization even after addition of 1 mM MnCl$_2$ (*Figure 2*, *Figure 4—figure supplement 1*). Additionally, substitution of the M2 coordinating residue D628 with alanine partially impaired dimerization in the presence of 1 mM MnCl$_2$, suggesting that manganese binding in the active site promoted dimerization (*Figure 4—figure supplement 1*).

To test whether manganese-induced dimerization correlated with phosphatase activation, we measured the dependence of phosphatase activity on manganese concentration using an assay for dephosphorylation of SpoIIAA-P, the native substrate of SpoIIE. By varying the manganese concentration in the presence of saturating substrate, we determined that SpoIIE was cooperatively activated ($h$ = 2.0) with a K$_{1/2}$ for manganese of 0.56 mM (*Figure 4C*). This correlates well with the manganese dependence of dimerization (*Figure 4B* left panel). Additionally, cooperative activation with a Hill coefficient of two indicates that at least two manganese ions bind in the active site of SpoIIE, consistent with the proposed catalytic mechanism (*Figure 3C*).

Our hypothesis also predicts that the V697A substitution would reduce the manganese concentration required for dimerization and phosphatase activity by favoring the active conformation of the switch. Indeed, the K$_{1/2}$ for manganese was reduced from 0.56 mM to 0.13 mM for SpoIIE[V697A] (*Figure 4C*), and SEC-MALLS revealed that the V697A substitution similarly reduced the concentrations of manganese required to promote dimer formation (*Figure 4B* right panel). Together these experiments provide biochemical evidence that SpoIIE dimerization is coupled to phosphatase activity by rotation of the switch region and coordination of manganese in the active site.

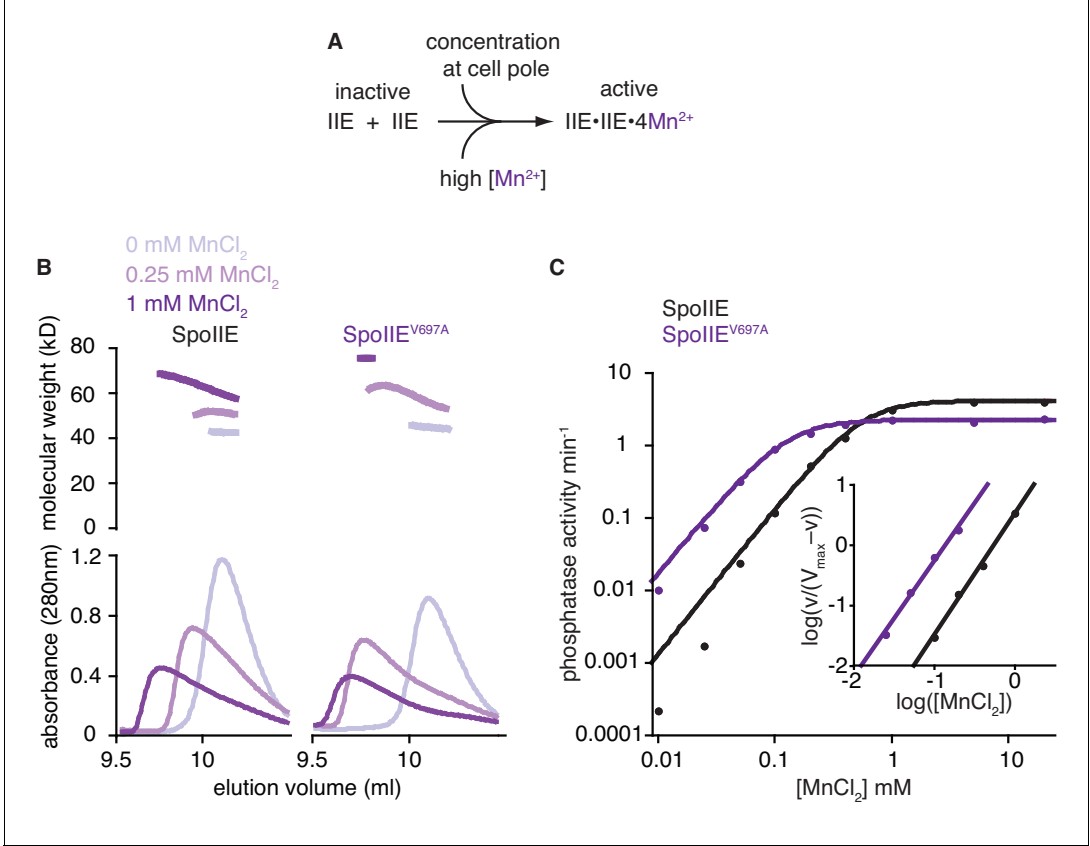

**Figure 4.** The switch promotes manganese binding in the phosphatase active site. **A** is a model for phosphatase activation. During sporulation, cellular cues induce dimerization of SpoIIE molecules, rotating the switch helices and leading to $Mn^{2+}$ binding in the active site. A prediction of this model is that high concentrations of $Mn^{2+}$ would drive SpoIIE to become activated and form dimers. **B** shows SEC-MALLS (size exclusion chromatography coupled to multi angle laser light scattering) results for the SpoIIE[457-827] fragment to assess complex formation at various concentrations of $Mn^{2+}$. The top plot shows molecular weights calculated from light scattering and the bottom plot shows the corresponding UV absorbance traces for both wild-type SpoIIE (left-hand side) and the gain-of-function mutant SpoIIE[V697A] (right). The experiments were performed in the absence of $Mn^{2+}$ (grey), with 0.25 mM $MnCl_2$ (light purple), and at 1 mM $MnCl_2$ (purple). All experiments were performed in triplicate and data from representative runs are shown. *Figure 4—figure supplement 1* shows size exclusion chromatography analysis of additional SpoIIE mutants. The source data are included as *Figure 4—source data 1*. **C** is a plot of phosphatase activity (initial rates, $v_{obs}$) for the wild-type (black) and V697A mutant (purple) SpoIIE[457–827] fragments as a function of $MnCl_2$ concentration using SpoIIAA-P as the substrate. The data were fit with the equation $v_{obs}=V_{max}*[MnCl_2]^h/(K+[MnCl_2]^h)$ where $h$ is the Hill coefficient calculated from the inset panel [$V_{max}$ = 4.15 ± 0.04 $min^{-1}$ (2.28 ± 0.04 $min^{-1}$ for SpoIIE[V697A]) and K = 0.32 ± 0.02 mM (0.020 ± 0.002 mM for SpoIIE[V697A])]. The $K_{1/2}$ values reported in the text were calculated from this equation and represent the concentration of $MnCl_2$ at which SpoIIE has half maximal activity. Inset is a Hill plot for data points representing 10–90% activity. Lines are linear fits to the data using the equation $log(v_{obs}/(V_{max}-v_{obs}))=h*log[MnCl_2]-logK$ [$h$ = 2.0 ± 0.1 (1.92 ± 0.1 for SpoIIE[V697A]) and K = 0.31 ± 0.04 mM (0.022 ± 0.008 mM for SpoIIE[V697A])]. The reported error is the error of the fit to the data. Experiments were repeated at least three times and data from a representative experiment are shown. The source data are included as *Figure 4—source data 1*.

The following source data and figure supplement are available for figure 4:

**Source data 1.** Source data for *Figure 4B and C*.

**Figure supplement 1.** Manganese-induced dimerization requires the active site and dimer interface.

# Discussion

We have presented the structures of the active and inactive state of the PP2C phosphatase SpoIIE from *B. subtilis*. Based on these structures, analysis of the function of SpoIIE mutants in vivo, and biochemical experiments, we propose that the movement of two helices at the base of the phosphatase

domain, forming the switch region, activates the phosphatase by positioning the carbonyl oxygen of a conserved glycine to coordinate manganese in the active site. Importantly, and as we will explain, structural and functional data additionally suggest that the switch mechanism is broadly conserved among PP2C phosphatases. Unexpectedly, and underscoring the flexibility and conservation of the switch, our analysis also reveals that a similar module controls the activity of proteases that form the catalytic core of the proteasome. This raises the possibility that the switch helices are a shared, and possibly evolutionarily conserved, feature of at least two families of enzymes that use unrelated catalytic mechanisms.

## The SpoIIE regulatory switch is broadly conserved among PP2C phosphatases

The following illustrative examples highlight the conservation and adaptability of the allosteric regulatory mechanism among PP2C phosphatases (*Figure 5*):

### RsbP

The phosphatase RsbP from *B. subtilis* is activated in response to energy stress by binding to its partner (RsbQ) to activate the transcription factor $\sigma^B$ (*Vijay et al., 2000*) (*Figure 5—figure supplement 1A*). Gain-of-function mutants of RsbP that constitutively activate $\sigma^B$ in the absence of RsbQ identified two elements in RsbP that control PP2C phosphatase activity (*Brody et al., 2009*). One element corresponds to one of the two switch helices we identified for SpoIIE ($\alpha$1). The other element (designated as $\alpha$0 by Brody et al.) was from the RsbP regulatory domain, and comparison with the structure of a closely related phosphatase (*Levchenko et al., 2009*) (RssB; the structure of RsbP itself has not been solved) suggests that this helix contacts the switch (*Figure 5—figure supplement 1B*). This suggests that RsbP and related phosphatases use the $\alpha$0 helix as a regulatory module to position the switch to control phosphatase activity. This supports the broad conservation of the switch mechanism and suggests that the switch is controlled by docking varied input domains with the switch helices.

### Pdp1

Pyruvate dehydrogenase phosphatase (Pdp1) dephosphorylates pyruvate dehydrogenase to promote respiratory metabolism (*Vassylyev and Symersky, 2007*) (*Figure 5—figure supplement 2A*). Human Pdp1 activity is inhibited by phosphorylation at a site distant from the active site (Y94), and phosphorylation at Y94 is commonly observed in human cancer cells and contributes to the Warburg effect (*Shan et al., 2014*). In the Pdp1 structure Y94 contacts a structural motif unique to Pdp1 (*Vassylyev and Symersky, 2007*) that packs against a pair of $\alpha$-helices structurally homologous to the SpoIIE switch helices (*Figure 5—figure supplement 2B*). We hypothesize that phosphorylation of Y94 would displace this structural element, shifting the position of the switch helices and inhibiting Pdp1 activity through a mechanism similar to that for SpoIIE regulation. Additionally, Pdp1 is activated by binding to the lipoyl moiety on the E2 subunit of the pyruvate dehydrogenase complex, and the proposed lipoic acid binding site is contained in the same structural element that is contacted by Y94 and packs against the switch helices (*Vassylyev and Symersky, 2007*) (*Figure 5—figure supplement 2B*). Thus, our model for PP2C regulation may explain how Pdp1 integrates both positive and negative regulatory signals to control phosphatase activity.

### Fem-2

The *C. elegans* Fem-2 phosphatase regulates sex determination in complex with its regulatory partners Fem-1 and Fem-3 (*Chin-Sang and Spence, 1996*) (*Figure 5—figure supplement 3A*). Additionally, the mammalian homologue of Fem-2 promotes caspase-dependent apoptosis by antagonizing $Ca^{2+}$/calmodulin-dependent protein kinase (*Tan et al., 2001*). Fem-2 has a specific N-terminal regulatory domain that is the scaffold for binding Fem-1 and Fem-3 to form the active complex (*Zhang et al., 2013*). In the Fem-2 structure this regulatory domain packs against the equivalent of the switch helices (*Zhang et al., 2013*) (*Figure 5—figure supplement 3B*). How Fem-2 phosphatase activity is regulated is not clear, but the direct contact between the Fem-2 regulatory domain and the switch helices is consistent with our proposed model for regulation of phosphatase activity through the switch helices.

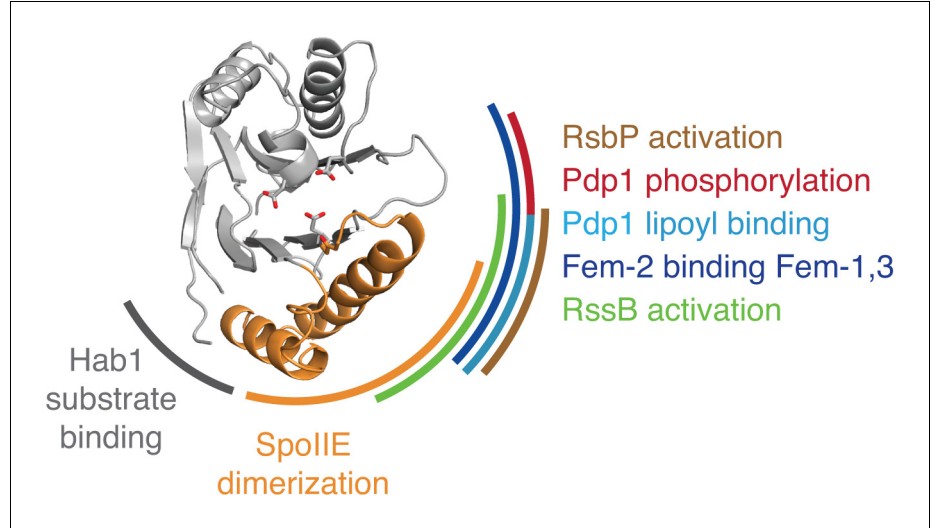

**Figure 5.** Evidence that the switch mechanism is broadly conserved among phosphatases. The structure of the active SpoIIE[457–827] phosphatase domain is shown in the center. The SpoIIE dimerization interface that mediates activation is indicated with an orange arc. Similarly, additional arcs indicate regions where regulatory inputs impinge on the PP2C phosphatase domain for RsbP (brown, *Figure 5—figure supplement 1*), Pdp1 (phosphorylation is shown in red, and lipoyl binding is shown in teal, *Figure 5—figure supplement 2*), Fem-2 (blue, *Figure 5—figure supplement 3*), Hab1 (grey, *Figure 5—figure supplement 4*), and RssB (green, *Figure 5—figure supplement 5*). The diagram is based on structures illustrated in *Figure 5—figure supplements 1–5*.

The following figure supplements are available for figure 5:

**Figure supplement 1.** Gain-of-function bypass suppressors suggest that the switch controls the energy stress response PP2C phosphatase RsbP.

**Figure supplement 2.** Structural evidence that the switch is used to control pyruvate dehydrogenase phosphatase.

**Figure supplement 3.** Structural evidence that the switch is used to control the sex-determining PP2C phosphatase FEM-2.

**Figure supplement 4.** The switch for the drought responsive PP2C phosphatase Hab1 switch could coordinate phosphatase activity and substrate binding.

**Figure supplement 5.** The switch for the pseudo-PP2C phosphatase RssB controls protease adapter activity.

## Hab1

The PP2C phosphatase Hab1 is a member of a sub-family of related phosphatases that regulate drought tolerance in response to abscisic acid in plants (*Ma et al., 2009*; *Park et al., 2009*) (*Figure 5—figure supplement 4A*). It is the only PP2C phosphatase for which a structure bound to its protein substrate (the kinase SnRK2) is available (*Soon et al., 2012*). Hab1 contacts SnRK2 primarily through a sub-domain (termed the 'flap' [*Pullen et al., 2004*]) that is variable in PP2C phosphatases and that packs against the switch helices (blue in *Figure 5—figure supplement 4B and C*) (*Soon et al., 2012*). In the presence of abscisic acid, the PYR/PYL/RCAR family of abscisic-acid-binding proteins inhibit Hab1 by binding to a site that overlaps with the binding site for SnRK2, suggesting that the switch could be influenced both by substrate and regulator binding. This could be a more general feature of PP2C phosphatases; the corresponding substrate-binding domain in SpoIIE changed conformation upon SpoIIE dimerization and activation (*Figure 5—figure supplement 4D*), suggesting that the conformational change of the switch could couple regulatory inputs to substrate

binding. Additionally, coupling between substrate binding and the active conformation of the switch helices would provide a conserved mechanism to achieve the known high substrate specificity of PP2C phosphatases.

### RssB

RssB activates the general stress response in *E. coli*, *P. aeruginosa*, and certain other gamma proteo-bacteria (*Battesti et al., 2011*). Although RssB is closely related to PP2C phosphatases, its primary role is not as a phosphatase (and does not require phosphatase activity), but rather as an adapter protein, delivering the transcription factor $\sigma^S$ for degradation by ClpXP in the absence of stress (*Figure 5—figure supplement 5A*). Structural and genetic studies revealed that adapter activity is regulated by contacts between the RssB regulatory domain and the switch helices that are mediated by dimerization, similar to our observations for SpoIIE (*Battesti et al., 2013*; *Levchenko et al., 2009*) (*Figure 5—figure supplement 5B and C*). Because the primary function of an adapter protein is to mediate protein-protein interactions, we hypothesize that for RssB the switch couples regulatory inputs to substrate binding (rather than to phosphatase activity) through a mechanism such as proposed above for Hab1. Thus, the switch mechanism may not only provide a flexible platform for adapting phosphatase activity to various inputs but also to control different outputs. We note that there are other pseudo-PP2C phosphatases including Tab1, which mediates caspase dependent apoptosis (*Lu et al., 2007*) and which may have similarly repurposed the switch mechanism.

Based on these examples, we conclude that the SpoIIE regulatory switch is broadly used to control diverse PP2C phosphatases via regulatory domains that dock on the switch to couple phosphatase activity to regulatory inputs (*Figure 5*).

## The PP2C switch is shared with the proteasome proteases

One of the most striking discoveries of our investigation is that the PP2C regulatory switch strongly resembles the allosteric switch that regulates the family of proteases that form the catalytic core of the proteasome (*Arciniega et al., 2014*; *Ruschak and Kay, 2012*; *Shi and Kay, 2014*; *Sousa et al., 2000*). These proteases are the most structurally similar family to PP2C phosphatases as revealed using the DALI server (*Holm and Rosenström, 2010*) and the ECOD database (*Cheng et al., 2014*), and like PP2C phosphatases their catalytic activity is subject to allosteric regulation. Specifically, the proteasome proteases and PP2C phosphatases have a conserved core fold (*Figure 6A and B*), which includes the switch helices, and the active sites are positioned in the same overall part of the structure. Although the proteases use different functional groups to mediate catalysis, the carbonyl oxygen of a conserved glycine (G629 of SpoIIE) at the junction of the core domain and the switch helices is used by both enzyme families for catalytic activity (*Sousa et al., 2000*).

Association with the regulatory cap activates the proteases, ensuring that the proteolytic active sites are sequestered prior to activation (*Seol et al., 1997*) (*Figure 6C*). Early studies on HslV, the *E. coli* homologue of the proteasome proteases, revealed that allosteric activation by the HslU cap takes place by rotation of the switch helices to position the active site glycine (*Figure 6D*) (*Sousa et al., 2000*). This mechanism is remarkably similar to the regulatory mechanism we proposed for PP2C phosphatases; docking of a regulatory module repositions the structurally homologous region in the same way to position the same functional group to achieve catalytic activity (*Video 2*).

This mechanism is also conserved for the archaeal proteasome, which like the eukaryotic proteasome includes an additional layer of related, but catalytically inactive $\alpha$ subunits; docking of the cap displaces the switch helices of the $\alpha$ subunits, which directly contact and reposition the switch helices of the catalytic $\beta$ subunits (*Ruschak and Kay, 2012*). Several lines of evidence suggest that this mechanism is conserved for the eukaryotic proteasome (*Arciniega et al., 2014*) and is additionally used by chaperones that promote proteasome maturation (*Wani et al., 2015*). Finally, comparative studies of the constitutive proteasome and the immune proteasome suggested that differences in the conformational flexibility of the switch underlies their differences in activity (*Arciniega et al., 2014*).

Thus, our identification of the PP2C switch demonstrates that PP2C phosphatases and the proteasome use the same allosteric regulatory module, revealing an unexpected link between two fundamental signaling systems – reversible phosphorylation and regulated proteolysis. Independent

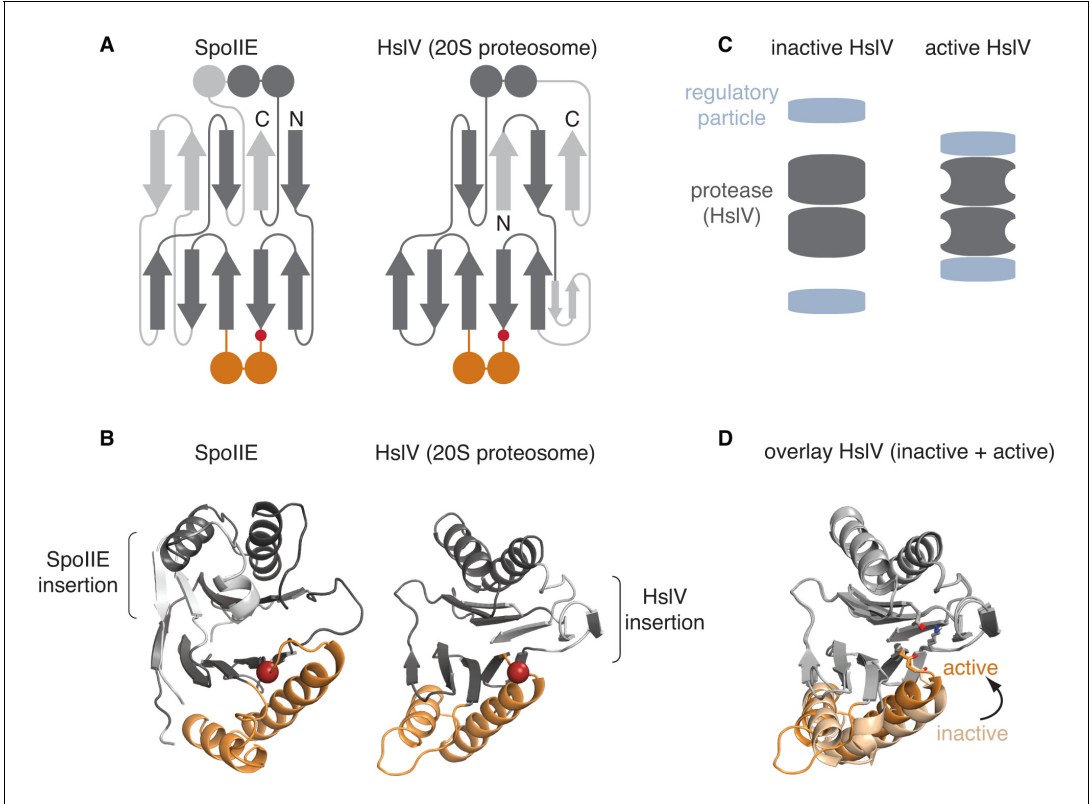

**Figure 6.** The switch mechanism is shared with proteasome proteases. **A** is a secondary structure topology diagram for SpoIIE (left) and for HslV (the *E. coli* homolog of the proteasome protease; right). *β* strands are shown as arrows pointing from N to C terminus and α-helices as circles in cross section. Conserved features are dark grey, whereas variable features are light grey. The conserved glycine that moves to activate each protein is indicated with a red circle. The switch helices of SpoIIE and the corresponding α-helices of HslV are colored orange. **B** shows ribbon diagrams of SpoIIE and HslV (PDB ID 1G3I) colored as in **A**. The position of the conserved regulatory glycine (G649 in SpoIIE, and G69 in HslV) is shown with a red sphere and the insertions specific to each protein are indicated by brackets. **C** is a schematic of how the regulatory particle (blue) activates the proteasome proteases (grey). **D** shows an overlay of the active (PDB ID 1G3I) and inactive (PDB ID 1G3K) states of HslV following superimposition of the regions in grey. The switch helices are color-coded orange and light orange for the active and inactive states, respectively. The active site residues T1, K33, and the carbonyl oxygen of G69 are shown.

analysis of structural and sequence similarity suggest that this is a result of common evolutionary ancestry. Structural comparison by the ECOD database, which classifies the evolutionary relationships of protein folds places PP2C phosphatases and the proteasome proteases in the same 'X-group', which is consistent with homology (*Cheng et al., 2014*). Independently, sequence-based searches using HHPRED (*Söding et al., 2005*) detected weak sequence similarity between phosphatases and the broad family of NTN-hydrolases that includes the proteasome proteases. For example, using the SpoIIE phosphatase domain sequence to search hidden Markov model alignments for *B. subtilis* proteins identified weak sequence similarity to D-fructose-6-phosphate amidotransferase, an NTN hydrolase. Notably, the region of possible sequence similarity maps to the switch helices and the *β* strands that follow and pack with the switch (although it is not known whether the switch helices play a regulatory role in amidotransferases).

## Allostery as a potential driver of evolutionary innovation

What evolutionary path might connect proteasomal proteases and PP2C phosphatases? Acquisition of a new catalytic mechanism requires that the ancestral protein retain function while acquiring the changes necessary for the new catalytic mechanism. However, conversion between the catalytic mechanisms of the proteasomal proteases and PP2C phosphatases would require multiple changes that would individually inactivate both activities (including circular permutation of the enzyme, loss/

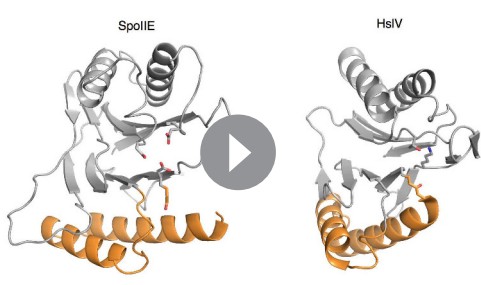

**Video 2.** PP2C phosphatases and proteasomal proteases share a common conformational switch. Shown are side-by-side displays of SpoIIE and HslV morphing from the inactive to active states. Shown on the left is the PP2C phosphatase domain of SpoIIE morphing from SpoIIE$^{590–827}$ (inactive, PDB ID 5MQH) to SpoIIE$^{457–827}$ (active, PDB ID 5UCG) as in *Figure 3B*. Shown on the right is HslV morphing from the HslU free structure (inactive, PDB ID 1G3K) to the HslU bound structure (active, PDB ID 1G3I) as in *Figure 6D*. The switch helices are colored orange and the active site residues of each protein are shown.

gain of metal binding, and charge swaps of residues at essential positions for catalysis). The conservation of the allosteric regulatory switch suggests a possible solution to this dilemma: namely, that the intermediate was a noncatalytic pseudoenzyme that retained the allosteric regulatory switch. RssB is an example of this sort of hypothetical pseudoenzyme; RssB uses the PP2C switch to regulate protease adapter function without functioning as a phosphatase (*Battesti et al., 2013*). An RssB-like intermediate would provide evolutionary pressure to preserve the regulatory mechanism, while creating a condition of neutrality to other mutations that would allow the new chemistry to evolve. Indeed, *E. coli* RssB lacks the C-terminal $\beta$ strand of PP2C phosphatases that is substituted by the N-terminus of the proteases (*Figure 6A*), suggesting a pathway for how a gene fusion event could produce the topological change required to evolve protease activity.

Allosteric regulatory modules have facilitated the evolutionary diversification of enzyme families to respond to new regulatory inputs, and the regulatory mechanism we have described for PP2C phosphatases may have similarly facilitated phosphatase diversification. A recent investigation of the evolution of ligand specificity in PDZ domains proposed that allostery produces conformational flexibility and thus may arise as a consequence of evolutionary history (*Raman et al., 2016*). Here we propose a mechanism whereby pre-existing allosteric regulatory modules such as we have identified for PP2C phosphatases facilitated the evolution of new enzymatic activities by transition through a pseudoenzyme intermediate that is pre-programmed for regulation. Pseudoenzymes are abundant (for example, 10% of kinase family members are pseudoenzymes) (*Leslie, 2013*) and thus may be important for their evolutionary potential in addition to their current biological functions.

## Materials and methods

### Construct design

The SpoIIE$^{457–827}$ construct was designed based on a putative sub-domain immediately N-terminal to the conserved PP2C phosphatase domain that we identified using HHPRED RRID:SCR_010276 (*Söding et al., 2005*). This region exhibited weak similarity to several proteins including another sporulation protein SpoIIIAH. Analysis of the regulatory domain (the newly determined portion of the structure) using the DALI server RRID:SCR_013433 (*Holm and Rosenström, 2010*) identified similarity to GpsL, a component of the type II secretion system, and structural alignment of the regulatory domain with SpoIIIAH matched the alignment predicted by HHPRED (*Söding et al., 2005*).

### Protein expression and purification

The SpoIIE$^{457–827}$ coding sequence was inserted into pET47b vector that had been digested with XmaI and XhoI using isothermal assembly. SpoIIE amino acid residue substitutions were introduced to this construct by Quikchange site directed mutagenesis. These constructs were introduced to *E. coli* BL21 (DE3) cells for protein expression. Cells were grown at room temperature to an OD$_{600}$ of 0.4, then were shifted to 14°C and expression was induced for 14–18 hr with 1 mM IPTG. Cells were harvested and pellets were resuspended in 5 ml/L of cell culture of 50 mM K•HEPES pH 8, 200 mM NaCl, 20 mM Imidazole, 10% Glycerol, 0.5 mM DTT, and 1 mM PMSF. Cells were lysed using a cell disruptor in one-shot mode (Constant Systems, Daventry, United Kingdom) and lysates were clarified by spinning for 30 min at 16,000 RPM in a Sorvall SS-34 rotor at 4°C. Lysates were loaded to a

HisTrap-HP column on an AKTA FPLC and eluted with a gradient of imidazole to 200 mM. The 6His tag was cleaved overnight with PreScission protease during dialysis to 50 mM K•HEPES pH 8, 200 mM NaCl, 20 mM Imidazole, 10% Glycerol, 0.5 mM DTT at 4°C. The PreScission protease was removed by flowing the dialyzed protein over a Ni-NTA resin, and the flowthrough was loaded to a Resource Q column that had been pre-equilibrated in 50 mM K•HEPES pH 8, 100 mM NaCl, 2 mM EDTA, 2 mM DTT. Protein was eluted using a gradient to 500 mM NaCl. Fractions containing SpoIIE were concentrated on Amicon Ultra centrifugal filters and loaded to a Superdex 200 column equilibrated with 20 mM K•HEPES pH 8, 50 mM NaCl, 2 mM DTT. Fractions containing SpoIIE were concentrated and immediately used to set up crystallization trials or were flash frozen in liquid nitrogen after addition of 10% glycerol.

Seleno-Methionine derivatized SpoIIE$^{457-827}$ protein was grown in fully supplemented M9 media. Fifteen minutes before induction, 100 mg/L L-Phenylalanine, 50 mg/L L-Isoleucine, 100 mg/L L-Lysine, 50 mg/L L-Leucine, 100 mg/L L-Threonine, 50 mg/L L-Valine, and 60 mg/L L-Selenomethionine were added. Otherwise induction and purification were identical to the un-derivatized protein.

Recombinant SpoIIE$^{590-827}$ (with an amino acid substitution A624I that was designed to block domain swapping) was overproduced from *E. coli* BL21 (DE3) harboring a pET-YSBLIC derivative plasmid. Cultures were grown at 37°C and induced at OD$_{600}$ = 0.6–0.7 by addition of IPTG to 1 mM followed by overnight growth at 16°C. Cells were harvested and the pellets resuspended in 20 mM sodium phosphate (pH 7.5), 0.5 M NaCl, 20 mM imidazole (Buffer A). The supernatant was loaded onto a HiTrap Ni-NTA column equilibrated with buffer A and eluted with a 20–500 mM imidazole gradient in buffer A. Fractions containing SpoIIE were concentrated before loading on to a Superdex S200 column equilibrated with 20 mM Tris pH 8.5, 150 mM NaCl.

## X-ray structure determination

SpoIIE$^{457-827}$ crystals were grown in sitting drops using Swissci 3 well 96 well plates (Hampton, Aliso Viejo, CA) with 40 µl well solution (0.5 mM LiSO$_4$, 8% PEG8000, 0.05 mM NaF, 6% glycerol). SpoIIE$^{457-827}$ (11 mg/mL) in 20 mM K•HEPES pH 8, 50 mM NaCl, 2 mM DTT was supplemented with 0.05 mM NaF and mixed at a 2:1 ratio with well solution in 300 nL drops using an NT8 robot (Formulatrix, Bedford, MA). Crystals grew over two weeks at room temperature. Crystals were cryo-protected by serial transfer to well solution supplemented with 10% and then 15% glycerol and plunged in liquid nitrogen. Data were collected at the Advanced Photon Source at Argonne National Laboratory on NE-CAT beamlines 24ID-C and 24ID-E.

Data were processed using HKL-2000 (*Otwinowski and Minor, 1997*) and initial phases were determined by molecular replacement using MR-PHASER RRID:SCR_014219 (*McCoy et al., 2007*) and an unswapped model from the published structure of SpoIIE$^{590-827}$ as the search model (*Levdikov et al., 2012*). Iterative model building and refinement was done in COOT RRID:SCR_ 014222 (*Emsley et al., 2010*) and refinement in PHENIX RRID:SCR_014224 (*Adams et al., 2010*). Non-crystallographic symmetry was initially enforced for the five chains in the asymmetric unit, then released first for chain B and finally for all chains. In later stages of refinement NCS was again imposed on regions where the chains differed by less than 4 Å. Model restraints were used based on the structure of SpoIIE$^{590-827}$ published here during an intermediate stage of refinement.

The model for SpoIIE$^{457-827}$ was additionally validated using anomalous signal from crystals grown with seleno-methionine derivatized protein (*Table 2*). With the exception of M557, signals were observed for all methionines at the expected sites in the anomalous difference map (an example is shown in *Figure 1—figure supplement 1A*).

Crystallization experiments with SpoIIE$^{590-827}$ consistently led to crystals of the domain-swapped dimeric form of the protein, even though SEC-MALLS analysis showed that SpoIIE$^{590-827}$ is predominantly monomeric (*Levdikov et al., 2012*). To stabilize the PP2C domain and slow down/prevent domain-swapping during crystallization, we introduced residue substitutions to reinforce the interface involved in domain-swapping. One such SpoIIE$^{590-827}$ mutant, A624I, constructed by Quikchange mutagenesis (changing the GCA codon to ATA), led to the crystallization of SpoIIE$^{590-827}$ without domain swapping. Residues with bulkier aliphatic side-chains (L, I, V or M) are found at the position corresponding to A624 in many SpoIIE orthologues.

Crystals of SpoIIE$^{590-827}$(A624I) were grown from hanging drops formed by mixing 1 µl of 38 mg/ mL protein with 1 µL of 2 M sodium formate, 100 mM sodium acetate, pH 4.6. The crystals were cryo-protected in mother liquor containing 4 M sodium formate for X-ray data collection on

**Table 3.** Table of strains. *B. subtilis* strains (all strains are in the background of PY79-RL3).

| Strain # | Genotype | Reference |
|---|---|---|
| RL3 | *prototrophic* | *Youngman et al., 1984* |
| RL5874 | *spoIIE::kan yxiD::spoIIE-yfp spc amyE::P$_{spoIIE}$-cfp cm* | *Bradshaw and Losick, 2015* |
| RL5902 | *spoIIE::kan yhdGH::P$_{spoIIQ}$-cfp tet amyE::spoIIE-yfp L646K spc* | *Bradshaw and Losick, 2015* |
| RL5904 | *spoIIE::kan yhdGH::P$_{spoIIQ}$-cfp tet amyE::spoIIE-yfp Q483A spc* | *Bradshaw and Losick, 2015* |
| RL5905 | *spoIIE::kan yhdGH::P$_{spoIIQ}$-cfp tet amyE::spoIIE-yfp G486K spc* | *Bradshaw and Losick, 2015* |
| RL5907 | *spoIIE::kan yhdGH::P$_{spoIIQ}$-cfp tet amyE::spoIIE-yfp E639K spc* | *Bradshaw and Losick, 2015* |
| RL6198 | *spoIIE::kan yhdGH::P$_{spoIIQ}$-cfp tet amyE::spoIIE-yfp V480K spc* | this study |
| RL6199 | *spoIIE::kan yhdGH::P$_{spoIIQ}$-cfp tet amyE::spoIIE-yfp A481K spc* | this study |
| RL6200 | *spoIIE::kan yhdGH::P$_{spoIIQ}$-cfp tet amyE::spoIIE-yfp L484K spc* | this study |
| RL6201 | *spoIIE::kan yhdGH::P$_{spoIIQ}$-cfp tet amyE::spoIIE-yfp V487K spc* | this study |
| RL6202 | *spoIIE::kan yhdGH::P$_{spoIIQ}$-cfp tet amyE::spoIIE-yfp S488K spc* | this study |
| RL6203 | *spoIIE::kan yhdGH::P$_{spoIIQ}$-cfp tet amyE::spoIIE-yfp V490K spc* | this study |
| RL6204 | *spoIIE::kan yhdGH::P$_{spoIIQ}$-cfp tet amyE::spoIIE-yfp M491K spc* | this study |
| RL6205 | *spoIIE::kan yhdGH::P$_{spoIIQ}$-cfp tet amyE::spoIIE-yfp D493K spc* | this study |
| RL6206 | *spoIIE::kan yhdGH::P$_{spoIIQ}$-cfp tet amyE::spoIIE-yfp F494K spc* | this study |
| RL6207 | *spoIIE::kan yhdGH::P$_{spoIIQ}$-cfp tet amyE::spoIIE-yfp S495K spc* | this study |
| RL6208 | *spoIIE::kan yhdGH::P$_{spoIIQ}$-cfp tet amyE::spoIIE-yfp E497K spc* | this study |
| RL6209 | *spoIIE::kan yhdGH::P$_{spoIIQ}$-cfp tet amyE::spoIIE-yfp I498K spc* | this study |
| RL6210 | *spoIIE::kan yhdGH::P$_{spoIIQ}$-cfp tet amyE::spoIIE-yfp E642K spc* | this study |
| RL6211 | *spoIIE::kan yhdGH::P$_{spoIIQ}$-cfp tet amyE::spoIIE-yfp I650K spc* | this study |
| RL6212 | *spoIIE::kan yhdGH::P$_{spoIIQ}$-cfp tet amyE::spoIIE-yfp T663K spc* | this study |
| RL6213 | *spoIIE::kan yhdGH::P$_{spoIIQ}$-cfp tet amyE::spoIIE-yfp I667K spc* | this study |
| RL5915 | *spoIIE::kan yhdGH::P$_{spoIIQ}$-cfp tet amyE::spoIIE-Δtag-yfp L646K spc* | *Bradshaw and Losick, 2015* |
| RL6246 | *spoIIE::kan yhdGH::P$_{spoIIQ}$-cfp tet amyE::spoIIE-Δtag-yfp V480K spc* | this study |
| RL6247 | *spoIIE::kan yhdGH::P$_{spoIIQ}$-cfp tet amyE::spoIIE-Δtag-yfp L484K spc* | this study |
| RL6248 | *spoIIE::kan yhdGH::P$_{spoIIQ}$-cfp tet amyE::spoIIE-Δtag-yfp V487K spc* | this study |
| RL6249 | *spoIIE::kan yhdGH::P$_{spoIIQ}$-cfp tet amyE::spoIIE-Δtag-yfp F494K spc* | this study |
| RL6250 | *spoIIE::kan yhdGH::P$_{spoIIQ}$-cfp tet amyE::spoIIE-Δtag-yfp I498K spc* | this study |
| RL6251 | *spoIIE::kan yhdGH::P$_{spoIIQ}$-cfp tet amyE::spoIIE-Δtag-yfp I650K spc* | this study |
| RL6252 | *spoIIE::kan yhdGH::P$_{spoIIQ}$-cfp tet amyE::spoIIE-Δtag-yfp T663K spc* | this study |
| RL6253 | *spoIIE::kan yhdGH::P$_{spoIIQ}$-cfp tet amyE::spoIIE-Δtag-yfp M491K spc* | this study |
| *E. coli* strains | | |
| RL6214 | *BL21 (DE3) pET47b H6-3C-spoIIE 457–827* | This study |
| RL6215 | *BL21 (DE3) pET47b H6-3C-spoIIE 457–827 V697A* | this study |
| RL6216 | *BL21 (DE3) pET47b H6-3C-spoIIE 457–827 D628A* | this study |
| RL6217 | *BL21 (DE3) pET47b H6-3C-spoIIE 457–827 L484K* | this study |
| RL6218 | *BL21 (DE3) pET23a H6-sumo-spoIIAA* | *Bradshaw and Losick, 2015* |
| RL6219 | *BL21 (DE3) pET23a H6-sumo-spoIIAB* | *Bradshaw and Losick, 2015* |
| AW2001 | *BL21 (DE3) pET-YSBLIC H6-3C-spoIIE 590–827 A624I* | *Levdikov et al., 2012* |
| AW2002 | *BL21 (DE3) pET-YSBLIC H6-3C-spoIIAA spoIIAB* | *Levdikov et al., 2012* |

beamline I02 at the DIAMOND Light Source. Data extending to 2.44 Å spacing were collected and processed using HKL-2000 (*Otwinowski and Minor, 1997*). Initial phases were determined by molecular replacement using MOLREP (*Vagin and Teplyakov, 2010*), and a coordinate set derived

from PDB ID 3T91 as the search model. The structure was rebuilt and refined using iterative cycles of COOT RRID:SCR_014222 (*Emsley et al., 2010*) and REFMAC RRID:SCR_014225 (*Murshudov et al., 1997*) respectively. Data collection and refinement statistics are given in *Table 1*.

## SEC-MALLS

SEC-MALLS was performed by loading 100 µL of 200 µM SpoIIE$^{457–827}$ to a Wyatt WTC-030S5 column using an Agilent HPLC in line with Wyatt DAWN-HELIOS and Optilab rEX detectors. Before running SpoIIE$^{457–827}$ was exchanged to 25 mM K•HEPES pH 8, 100 mM NaCl using a Superdex 200 column. The SEC-MALLS instrument was equilibrated in 25 mM K•HEPES pH 8, 100 mM NaCl, supplemented with MnCl$_2$ as appropriate. SpoIIE$^{457–827}$ samples were supplemented with MnCl$_2$ shortly before running on the SEC-MALLS. Analysis was conducted using the ASTRA software. All SEC-MALLS samples were run in at least triplicate. SEC experiments shown in *Figure 4—figure supplement 1* were conducted similarly, loading 200 µL of 200 µM SpoIIE$^{457–827}$ on a 20 mL Superdex 200 column on an AKTA FPLC.

## Phosphatase assays

Phosphatase assays were performed as reported in *Bradshaw and Losick, 2015*. SpoIIAA, SpoIIAA-P, and SpoIIAB were produced and purified as described previously. SpoIIAA-P was produced by overexpression of 6H-SpoIIAA in an *E. coli* strain that also expressed SpoIIAB (*Levdikov et al., 2012*). To produce $^{32}$P labeled SpoIIAA-P, 75 µM SpoIIAA, 5 µM SpoIIAB and 50 µCi of γ-$^{32}$P ATP were incubated overnight in 50 mM K•HEPES pH 7.5, 50 mM KCl, 750 µM MgCl$_2$, 2 mM DTT. The protein was exchanged to 20 mM K•HEPES pH 7.5, 200 mM NaCl, 2 mM DTT over a Zeba spin column (Pierce) to remove unincorporated nucleotide and then flowed over Q sepharose resin to remove SpoIIAB. Phosphatase assays were performed in 25 mM K•HEPES pH 8, 100 mM NaCl, 100 µg/ml BSA (supplemented with MnCl$_2$ as appropriate) with 2.5 µM SpoIIE and 200 µM SpoIIAA-P. Reactions were started by adding SpoIIE to a mixture containing SpoIIAA-P and MnCl$_2$. Reactions were stopped in 1 M KPO$_4$ pH 3.3, 2% Triton X-100 and run on PEI-Cellulose TLC plates developed in 1 M LiCl$_2$, 0.8 M Acetic Acid, and imaged on a Typhoon (GE Life Sciences, Pittsburgh, PA). Phosphatase assays were performed more than three independent times as separate experiments.

## *B. subtilis* strains and analysis

*B. subtilis* strains were constructed using standard molecular genetic techniques (*Harwood and Cutting, 2010*) in the PY79 strain background (*Youngman et al., 1984*; *Zeigler et al., 2008*) and were validated to contain the correct constructs by double-crossover recombination at the correct insertion site. All strains used in this study are described in *Table 3*. For imaging, cells were grown at 37°C in 25% LB to OD 0.6, resuspended in minimal sporulation resuspension medium, and grown for 2.5 hr. Cells were immobilized on 2.5% agarose pads made with the sporulation resuspension medium and imaged on an Olympus BX-61 upright microscope with a 100X objective. Cells were segmented using SuperSegger (*Stylianidou et al., 2016*) and analyzed with custom MatLab scripts (*Bradshaw and Losick, 2015*). Samples were taken from the same cultures for western blot analysis; cells were lysed using a FastPrep (MP-BIO, Santa Ana, CA) and blots were probed with polyclonal α-GFP antibody.

## Acknowledgements

We thank E Blagova, J Turkenburg, E Dodson, J Tunaley, R Grant and the beam staff at NE-CAT and the Diamond Light Source for expert assistance, and T Baker and R Grant for access to a home X-ray source and the SEC-MALLS instrument. We also thank S McKnight, L Shapiro, K Ramamurthi, C Price, J Kardon, M Cabeen, S Wacker, and J Russell for valuable comments during manuscript preparation. X-ray data were collected using NE-CAT beamlines (GM103403) and a Pilatus detector (RR029205) at the APS (DE-AC02-06CH11357), and beamline I02 (mx-7864) at the Diamond Light Source. Coordinates are deposited at the PDB with accession numbers (5UCG and 5MQH)

## Additional information

### Funding

| Funder | Grant reference number | Author |
|---|---|---|
| National Institutes of Health | GM18568 | Richard Losick |
| Wellcome | 082829 | Anthony J Wilkinson |
| Damon Runyon Cancer Research Foundation | DRG 2051-10 | Niels Bradshaw |
| Jane Coffin Childs Memorial Fund for Medical Research | | Christina M Zimanyi |

The funders had no role in study design, data collection and interpretation, or the decision to submit the work for publication.

### Author contributions

NB, Conceptualization, Investigation, Visualization, Writing—original draft, Writing—review and editing; VML, Conceptualization, Investigation, Writing—review and editing; CMZ, Investigation, Writing—review and editing; RG, Supervision, Writing—review and editing; AJW, Conceptualization, Funding acquisition, Writing—review and editing; RL, Conceptualization, Funding acquisition, Writing—original draft, Writing—review and editing

### Author ORCIDs

Niels Bradshaw, http://orcid.org/0000-0002-6845-4717
Christina M Zimanyi, http://orcid.org/0000-0002-6782-507X
Rachelle Gaudet, http://orcid.org/0000-0002-9177-054X
Anthony J Wilkinson, http://orcid.org/0000-0003-4577-9479
Richard Losick, http://orcid.org/0000-0002-5130-6582

## Additional files

### Major datasets

The following datasets were generated:

| Author(s) | Year | Dataset title | Dataset URL | Database, license, and accessibility information |
|---|---|---|---|---|
| Bradshaw N, Levdikov VM, Zimanyi CM, Gaudet R, Wilkinson AJ, Losick R | 2016 | Structure of the PP2C Phosphatase Domain and a Fragment of the Regulatory Domain of the Cell Fate Determinant SpoIIE from Bacillus Subtilis | http://www.rcsb.org/pdb/explore/explore.do?structureId=5UCG | Publicly available at the RCSB Protein Data Bank (accession no: 5UCG) |
| Levdikov VM, Wilkinson AJ, Blagova EV | 2016 | Structure of the Phosphatase Domain of the Cell Fate Determinant SpoIIE from Bacillus subtilis in a crystal form without domain swapping | http://www.rcsb.org/pdb/explore/explore.do?structureId=5MQH | Publicly available at the RCSB Protein Data Bank (accession no: 5MQH) |

The following previously published datasets were used:

| Author(s) | Year | Dataset title | Dataset URL | Database, license, and accessibility information |
|---|---|---|---|---|
| Levdikov VM, Wilkinson AJ, Blagova EV | 2011 | Structure of the Phosphatase Domain of the Cell Fate Determinant SpoIIE from Bacillus subtilis | http://www.rcsb.org/pdb/explore/explore.do?structureId=3t91 | Publicly available at the RCSB Protein Data Bank (accession no: 3T91) |
| Sousa MC, Trame CB, Tsuruta H, Wilbanks SM, Red- | 2000 | CRYSTAL STRUCTURE OF THE HSLUV PROTEASE-CHAPERONE COMPLEX | http://www.rcsb.org/pdb/explore/explore.do?structureId=1g3i | Publicly available at the RCSB Protein Data Bank (accession no: |

| | | | | |
|---|---|---|---|---|
| dy VS, McKay DB | | | | 1G3I) |
| Sousa MC, McKay DB | 2000 | CRYSTAL STRUCTURE OF THE H. INFLUENZAE PROTEASE HSLV AT 1.9 A RESOLUTION | http://www.rcsb.org/pdb/explore/explore.do?structureId=1g3k | Publicly available at the RCSB Protein Data Bank (accession no: 1G3K) |
| Teh AH, Makino M, Baba S, Shimizu N, Yamamoto M, Ku-masaka T | 2013 | Crystal structure of RsbX in complex with manganese in space group P21 | http://www.rcsb.org/pdb/explore/explore.do?structureId=3w43 | Publicly available at the RCSB Protein Data Bank (accession no: 3W43) |
| Levchenko I, Grant RA, Sauer RT, Baker TA | 2008 | Structure of Orthorhombic crystal form of Pseudomonas aeruginosa RssB | http://www.rcsb.org/pdb/explore/explore.do?structureId=3f7a | Publicly available at the RCSB Protein Data Bank (accession no: 3F7A) |
| Vassylyev DG, Sy-mersky J | 2007 | Crystal structure of pyruvate dehydrogenase phosphatase 1 (PDP1) | http://www.rcsb.org/pdb/explore/explore.do?structureId=2pnq | Publicly available at the RCSB Protein Data Bank (accession no: 2PNQ) |
| Feng Y, Zhang Y, Ge J, Yang M | 2013 | Structure of a C.elegans sex determining protein | http://www.rcsb.org/pdb/explore/explore.do?structureId=4jnd | Publicly available at the RCSB Protein Data Bank (accession no: 4JND) |
| Zhou XE, Soon, F-F, Ng L-M, Kovach A, Tan MHE, Suino-Powell KM, He Y, Xu Y, Brunzelle JS, Li J, Melcher K, Xu HE | 2011 | Crystal structure of SnRK2.6 in complex with HAB1 | http://www.rcsb.org/pdb/explore/explore.do?structureId=3ujg | Publicly available at the RCSB Protein Data Bank (accession no: 3UJG) |

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
