## [Decision Letter]

Thank you for submitting your article "PP2C phosphatases share a common regulatory switch with proteasomal proteases" for consideration by *eLife*. Your article has been favorably evaluated by Gisela Storz (Senior Editor) and three reviewers, one of whom, Michael Laub, is a member of our Board of Reviewing Editors.

The reviewers have discussed the reviews with one another and the Reviewing Editor has drafted this decision to help you prepare a revised submission.

Summary:

This paper from Bradshaw, Losick and colleagues presents evidence that the PP2C-like phosphatase SpoIIE in *B. subtilis* undergoes a conformational change upon activation that involves dimerization and rotation of two key helices, which promote binding of the divalent ions crucial for catalysis. The structures are compelling and highly suggestive of the model proposed, with the biochemical and genetic evidence further substantiating the model. Notably, the authors synthesize their findings with the structures of several other PP2C family proteins to suggest that a similar switch is used throughout this family of proteins, including in proteasomal proteases. Although the reviewers were each enthusiastic about the paper and the identification of a common regulatory switch mechanism, some aspects of the paper could be improved.

Essential revisions:

The data in Figure 2—figure supplement 1 could have been better presented, in the figure and the text. For one, the wild-type profile is shown as a gray line, but the corresponding image is not shown and would be helpful to the uninitiated reader. Additionally, there was confusion about the claim that "The results revealed that a continuous region of the dimer interface composed of six residues from the long α-helix of the regulatory domain and two residues from the switch helices were needed for all three aspects of SpoIIE function." It's not clear which 8 residues the authors are referring to here. Figure 2 highlights 12 positions in total, Figure 2 highlights 10 residues at the interface, and Figure 2—figure supplement 1 shows data for 16 mutants all of which seem defective to some extent in at least one of the assays shown. Some confusion also arises from an apparent error in the text in the subsection “Amino acid substitutions in the dimer interface block function” – "two" should be "three" (i.e., E639, E642, and I667), as stated in the Figure 2 legend. In short, a significant improvement in clarity is needed – it almost seemed as if the authors were trying to be so succinct here that they made this section rather opaque. Also, perhaps most importantly, it's not clear one can argue that 'all three aspects' of function (see a similar statement in the subsection “Repositioning the switch region is necessary for phosphatase activation”) are perturbed in these mutants if one of those functions is stability, i.e. if it's not stable, you can't assess phosphatase activity or localization since there's no protein present! One would need to stabilize the unstable mutants another way, e.g. by mutating the protease, and then assess phosphatase activity and localization.

The results of Figure 2—figure supplement 1 seem so critical to the paper's conclusions that they really should be in a main figure.

The authors seem to be assuming that mutations affecting function must be eliminating dimerization, but there's no direct assay of dimerization for most mutants, except one in Figure 4—figure supplement 1 (L484K). And the other, D628A, doesn't seem as defective in dimer formation, in conflict with the claim at the end of the first paragraph of the subsection “Mn^2+^ stimulates dimerization and phosphatase activity”.

If, as the authors argue in the first paragraph of the subsection “Evidence from gain-of-function mutants that repositioning the switch helices is sufficient for phosphatase activation”, V697 plays a crucial role in stabilizing the inactive state, then it seems like replacing it with residues other than just alanine, e.g. charged or bulky polar residues, should also destabilize the inactive state, yielding hyperactivity. Has this been, or can this be, tested? For all of the gain-of-function mutants, the prediction is that they are constitutive dimers – can this be tested biochemically, e.g. using the SEC-MALLS assay in Figure 4? Or do any of the gain-of-function mutations activate without dimerization?

The structure said to represent the active form probably lacks the second manganese ion at the active site. The anomalous difference map in Figure 3—figure supplement 3 suggests only one manganese ion is present. The authors state that "the number of bound metal atoms could not be established", but fail to note the apparent absence of the second manganese ion. Was the protein analyzed by atomic absorption spectroscopy? Also, regarding the number of Mn atoms indicated by the Mn anomalous difference map: because the resolution was so low (5.4 Å), it's not impossible that two atoms (or one atom) could appear as one electron density blob. The authors could expand their discussion here and perhaps include the density in their alignment in Figure 3—figure supplement 3 panel C.

Related to the previous comment, did the authors try to crystallize or quantify manganese for SpoIIE 457-827 with a substitution that constitutively activates?

In Figure 2—figure supplement 1, 5 of 7 mutants in the righthand column appear to mislocalize SpoIIE closer to the forespore pole, which the authors ignore. Can the authors offer an explanation?

It is stated in the Figure 1 panel D legend that there are two and a half dimers in the asymmetric unit. Does aligning the five SpoIIE models reveal structural differences?

The authors seem to imply that the SpoIIE domain-swap is a crystallization artifact. Briefly explain the arguments for or against the biological relevance of the domain-swapped dimer.

---

## [Author Response]

*Summary:*

*This paper from Bradshaw, Losick and colleagues presents evidence that the PP2C-like phosphatase SpoIIE in B. subtilis undergoes a conformational change upon activation that involves dimerization and rotation of two key helices, which promote binding of the divalent ions crucial for catalysis. The structures are compelling and highly suggestive of the model proposed, with the biochemical and genetic evidence further substantiating the model. Notably, the authors synthesize their findings with the structures of several other PP2C family proteins to suggest that a similar switch is used throughout this family of proteins, including in proteasomal proteases. Although the reviewers were each enthusiastic about the paper and the identification of a common regulatory switch mechanism, some aspects of the paper could be improved.*

In addition to the changes listed below that directly respond to the reviewers’ comments, we have made a few changes to strengthen the manuscript:

We additionally refined the model for the SpoIIE^457-827^ structure. The statistics for the structure improved slightly and have been updated in Table 1. Most of the changes were subtle adjustments of sidechains, but we were able to build the “flap” region for a few of the chains and this is now reflected in Figure 5—figure supplement 4.We modified the title to make it accessible to a broader audience (we dropped PP2C).We modified the discussion of the evolutionary implications of our findings to emphasize the incompatibility of the catalytic mechanisms of PP2C phosphatases and the proteasomal proteases.We have added two supplementary videos (morphs) that illustrate the allosteric regulatory mechanisms of SpoIIE and HslV.

*Essential revisions:*

*The data in Figure 2—figure supplement 1 could have been better presented, in the figure and the text. For one, the wild-type profile is shown as a gray line, but the corresponding image is not shown and would be helpful to the uninitiated reader.*

We have added an image of a representative wild type cell to the figure.

*Additionally, there was confusion about the claim that "The results revealed that a continuous region of the dimer interface composed of six residues from the long α-helix of the regulatory domain and two residues from the switch helices were needed for all three aspects of SpoIIE function." It's not clear which 8 residues the authors are referring to here. Figure 2 highlights 12 positions in total, Figure 2 highlights 10 residues at the interface, and Figure 2—figure supplement 1 shows data for 16 mutants all of which seem defective to some extent in at least one of the assays shown. Some confusion also arises from an apparent error in the text in the subsection “Amino acid substitutions in the dimer interface block function” – "two" should be "three" (i.e., E639, E642, and I667), as stated in the Figure 2 legend. In short, a significant improvement in clarity is needed – it almost seemed as if the authors were trying to be so succinct here that they made this section rather opaque.*

We have added text to specify explicitly the residues we are referring to: the six residues from the long α-helix of the regulatory domain are V480, L484, V487, M491, F494, and I498 and the three residues from the switch helices L646, I650, and T663 (subsection “Amino acid substitutions in the dimer interface block function”).

*Also, perhaps most importantly, it's not clear one can argue that 'all three aspects' of function (see a similar statement in the subsection “Repositioning the switch region is necessary for phosphatase activation”) are perturbed in these mutants if one of those functions is stability, i.e. if it's not stable, you can't assess phosphatase activity or localization since there's no protein present! One would need to stabilize the unstable mutants another way, e.g. by mutating the protease, and then assess phosphatase activity and localization.*

Good point! We had only done this for one of the mutants previously (L646K). We have now deleted the FtsH degradation tag from the nine variants of SpoIIE that were destabilized and assayed SpoIIE protein levels and σ^F^ activation during sporulation. We found that all nine variants accumulated to levels substantially in excess of SpoIIE containing the FtsH degradation tag but nonetheless all variants were defective for σ^F^ activation. We have added these data to Figure 2—figure supplement 1 as panel C.

*The results of Figure 2—figure supplement 1 seem so critical to the paper's conclusions that they really should be in a main figure.*

Because Figure 2 summarizes the results presented in Figure 2—figure supplement 1, we feel that it works best to have this figure as a supplement. The *eLife* online format in which supplementary figures are made easily accessible by clicking on the main figure is particularly suited to making the underlying data summarized in a main figure easily available. However, if the reviewers feel strongly about this, we would be willing to include it as a main figure.

*The authors seem to be assuming that mutations affecting function must be eliminating dimerization, but there's no direct assay of dimerization for most mutants, except one in Figure 4—figure supplement 1 (L484K).*

We reasoned that substitution of the native amino acids in the dimer interface with lysine would disrupt the dimer interface because of the introduction of a bulky amino acid with a positive charge. We have added text stating, “We substituted the native amino acids with lysine because the introduced charge and the long sidechain would be expected to prevent dimerization” to clarify our reasoning. Consistent with our reasoning, the positions of amino-acid substitutions that produced phenotypes are clustered in the dimer interface, and the variant we tested in vitro (L484K) blocked dimerization.

*And the other, D628A, doesn't seem as defective in dimer formation, in conflict with the claim at the end of the first paragraph of the subsection “Mn^2+^ stimulates dimerization and phosphatase activity”.*

We have modified the text to indicate that the D628A substitution partially blocks manganese induced dimer formation. The fact that some dimerization still occurs for the D628A variant is not surprising because the other metal coordinating residues are still present (including G629) and high concentrations of protein and MnCl_2_ were used in the experiment (text has been added to the legend of Figure 4—figure supplement 1).

*If, as the authors argue in the first paragraph of the subsection “Evidence from gain-of-function mutants that repositioning the switch helices is sufficient for phosphatase activation”, V697 plays a crucial role in stabilizing the inactive state, then it seems like replacing it with residues other than just alanine, e.g. charged or bulky polar residues, should also destabilize the inactive state, yielding hyperactivity. Has this been, or can this be, tested?*

Yes. We have isolated an additional suppressor allele at this position (V697F) that hyperactivates SpoIIE, consistent with the reviewer’s reasoning. We have included this in the text (subsection “Structure of the phosphatase domain”).

*For all of the gain-of-function mutants, the prediction is that they are constitutive dimers – can this be tested biochemically, e.g. using the SEC-MALLS assay in Figure 4?*

We focused on the V697A variant because it has several distinguishing characteristics: (1) V697A is the only variant that we have isolated in the crystalized fragment that led to hyperactivity of SpoIIE in the otherwise wild-type context (the gain-of-function mutants were isolated as suppressors of defective alleles of SpoIIE) and (2) V697A substitution restores phosphatase activity to all of the SpoIIE alleles we have tested that disrupt SpoIIE activation (with the exception of alleles that disrupt the catalytic center). We have not tested the other variants in vitro, but suspect that some may have similar albeit milder effects.

*Or do any of the gain-of-function mutations activate without dimerization?*

Based on unpublished data, we have evidence that the V697A substitution can promote SpoIIE activity independent from dimerization (as would be predicted from our model for SpoIIE activation). Specifically, expression of the PP2C domain fragment SpoIIE^590-827^ in vivo fails to support σ^F^ activation, but σ^F^ activation can be partially rescued by V697A substitution in the SpoIIE^590-827^ construct. Since this fragment lacks the regulatory domain that templates dimerization, this supports the hypothesis that V697A can promote SpoIIE activity in the absence of dimerization.

*The structure said to represent the active form probably lacks the second manganese ion at the active site. The anomalous difference map in Figure 3—figure supplement 3 suggests only one manganese ion is present. The authors state that "the number of bound metal atoms could not be established", but fail to note the apparent absence of the second manganese ion. Was the protein analyzed by atomic absorption spectroscopy? Also, regarding the number of Mn atoms indicated by the Mn anomalous difference map: because the resolution was so low (5.4* Å*), it's not impossible that two atoms (or one atom) could appear as one electron density blob. The authors could expand their discussion here and perhaps include the density in their alignment in Figure 3—figure supplement 3 panel C.*

As noted, SpoIIE^457-827^ did not contain manganese under the conditions we used for crystallization and determination of the SpoIIE^457-827^ structure (we confirmed that manganese was absent from the crystals using X-ray fluorescence scans). We agree that the low resolution anomalous maps were not conclusive with respect to the number of manganese ions bound in the structure (or the site of the bound manganese) and have modified the text to clarify this point (subsection “The switch helices move a conserved manganese-coordinating residue into the active site”, last paragraph). The density for chain A was shown in Figure 3—figure supplement 4 panel A and we have added density maps for the active sites of the other four chains to this figure panel. We have not analyzed the protein by atomic absorption spectroscopy because of the low affinity of manganese binding, however we present new biochemical evidence that SpoIIE uses two manganese ions in the catalytic site in Figure 4. By extending the enzymatic analysis of SpoIIE phosphatase activity with better data at low concentrations of manganese, we found that SpoIIE is cooperatively activated by manganese with a Hill coefficient of two, indicating that at least two manganese ions are required for catalysis, consistent with the proposed model for SpoIIE catalytic activity. We have added discussion of this point to the manuscript (subsection “Mn^2+^ stimulates dimerization and phosphatase activity”, second paragraph).

*Related to the previous comment, did the authors try to crystallize or quantify manganese for SpoIIE 457-827 with a substitution that constitutively activates?*

We have not yet attempted to crystalize the SpoIIE^457-827^ fragment with the V697A substitution, but agree that this might facilitate getting a high resolution structure bound to manganese.

*In Figure 2—figure supplement 1, 5 of 7 mutants in the righthand column appear to mislocalize SpoIIE closer to the forespore pole, which the authors ignore. Can the authors offer an explanation?*

We have added text to the legend of Figure 2—figure supplement 1 that clarifies this point: “After σ^F^ activation, SpoIIE is recruited back to the forespore face of the asymmetric septum and then moves along with the engulfing membrane to encompass the forespore, thus mutants with the most severe defects in σ^F^ activation have fluorescence profiles that are slightly shifted towards the forespore pole relative to wild-type cells.”

*It is stated in the Figure 1 panel D legend that there are two and a half dimers in the asymmetric unit. Does aligning the five SpoIIE models reveal structural differences?*

There are subtle differences between the chains that can be detected at the resolution of our data, but they are mostly limited to the regions of crystal contacts. Some of the chains have density in the “flap” region that we have now built and added to Figure 5—figure supplement 4.

*The authors seem to imply that the SpoIIE domain-swap is a crystallization artifact. Briefly explain the arguments for or against the biological relevance of the domain-swapped dimer.*

Solution studies of the SpoIIE^590-827^ fragment established that it is monomeric under physiological conditions and can form domain swapped dimers in crystallization conditions (this is discussed in more detail in Levdikov et al. 2012). Additionally, our structure of the A624I mutant revealed that the major conformational differences between the domain swapped structure and the dimeric structure are not caused by the domain swap. Finally, bulkier aliphatic side-chains are found at the position corresponding to A624 in SpoIIE orthologues. Although we cannot rule out the possibility that the domain swapped dimer is physiologically relevant, we do not favor it for these reasons.